# A spatio-temporal brain miRNA expression atlas identifies sex-independent age-related microglial driven miR-155-5p increase

Annika Engel [1,9], Viktoria Wagner[1,2,9], Oliver Hahn[2,8], Aulden G. Foltz [2], Micaiah Atkins [2], Amila Beganovic[1], Ian H. Guldner[2,3], Nannan Lu [2,3], Aryaman Saksena[2], Ulrike Fischer [4], Nicole Ludwig[1,4], Eckart Meese[4], Tony Wyss-Coray [2,3,5] & Andreas Keller [1,6,7] ✉

An in-depth understanding of the molecular processes composing aging is crucial to develop therapeutic approaches that decrease aging as a key risk factor for cognitive decline. Herein, we present a spatio-temporal brain atlas (15 different regions) of microRNA expression across the mouse lifespan (7 time points) and two aging interventions. MicroRNAs are promising therapeutic targets, as they silence genes by complementary base-pair binding of messenger RNAs and mediate aging speed. We first established sex- and brain-region-specific microRNA expression patterns in young adult samples. Then we focused on sex-dependent and independent brain-region-specific microRNA expression changes during aging. We identified three sex-independent brain aging microRNAs (miR-146a-5p, miR-155-5p, and miR-5100). For miR-155-5p, we showed that these expression changes are driven by aging microglia and target mTOR signaling pathway components and other cellular communication pathways. In this work, we identify strong sex-brain-region-specific aging microRNAs and microglial miR-155-5p as a promising therapeutic target.

Aging, which is defined as the loss of physiological functions resulting in the end of lifespan, impacts all organs[1]. But the mechanisms driving the gradual decline in the brain and the resulting loss of its functionalities are yet to be understood[2]. Molecular and neuroimaging studies suggested that regions of the brain may experience varying degrees of susceptibility to the effects of the aging process[3,4]. Brain aging atlases of epigenetic modifications[5] and transcriptional changes[3] exist, but the miRNA expression layer is missing. Mature miRNAs are short (18–24 nt) single-stranded RNA molecules that play an important role in post-transcriptional regulation via complementary base pair binding to mRNA molecules[6], thereby reducing their stability and inhibiting

translation[7]. Their capacities of targeting multiple mRNAs in the same pathways[8] and their stability make them promising targets for therapy applications and diagnostics[9]. MiRNAs mediate protective microglial states in an AD mouse model[10], were proposed as important drivers of aging-related phenotypes cross-organs[11] and shown to control aging speed via exosomes[12]. Hence, generating a comprehensive atlas of miRNA expression during healthy brain aging is urgently needed to characterize how and where aging occurs and leads to vulnerabilities, such as neurological disorders like Alzheimer's (AD) or Parkinson's disease (PD)[13–17]. Currently, information on miRNA expression in different brain regions is rather sparse[18–20]. Many studies focus on a set of

[1]Clinical Bioinformatics, Saarland University, Saarbrücken, Germany. [2]Department of Neurology and Neurological Sciences, Stanford University, Stanford, CA, USA. [3]Wu Tsai Neurosciences Institute, Stanford University School of Medicine, Stanford, CA, USA. [4]Department of Human Genetics, Saarland University, Homburg/Saar, Germany. [5]The Phil and Penny Knight Initiative for Brain Resilience, Stanford University, Stanford, CA, USA. [6]Helmholtz Institute for Pharmaceutical Research Saarland, Helmholtz Center for Infection Research, Saarbrücken, Germany. [7]PharmaScienceHub, Saarland University, Saarbrücken, Germany. [8]Present address: Calico Life Sciences LLC, San Francisco, CA, USA. [9]These authors contributed equally: Annika Engel, Viktoria Wagner. ✉e-mail: andreas.keller@ccb.uni-saarland.de

2–12 regions and predominantly used male mice with few to no differing age groups[18,20]. Through NGS, we were able to assess another layer of miRNA biology, previously unexplored in the aging context, the isomiR expression. IsomiRs are naturally occurring clinically relevant variants of the mature archetype miRNA[21].

We decided to use inbred mouse samples to minimize genetic and environmental variability, creating a baseline of miRNA expression with minimal confounding factors. A baseline is essential for human studies as these are challenging due to the sparse availability of unaffected brains, genetic and environmental heterogeneity, varying post-mortem intervals, and hence varying sample quality.

In this work, we analyze region-specific miRNA expression patterns in males and females separately to uncover sex-specific regional expression. Building on the sex-specific analysis, we study age-related miRNA expression changes in a sex-dependent and independent manner. We identify three cross-sex brain aging miRNAs, likely derived from microglial expression changes. We check the expression changes of our brain aging miRNAs in two aging interventions, dietary restriction and young plasma injection. Finally, we combined our miRNA dataset with the existing mRNA dataset and found that cellular communication and mTOR signaling pathways were regulated by brain aging miRNA miR-155-5p.

## Results

### Brain region-specific miRNA expression patterns

Aging is the main risk factor to suffer from major neurodegenerative diseases, therefore, understanding the underlying mechanisms is crucial for the development of effective therapies[13]. We generated bulk sequencing data from 844 samples from 15 defined regions at seven ages (3, 12, 15, 18, 21, 26, 28 months) to uncover the region-specific miRNA patterns during aging. Samples from the following regions were collected: corpus callosum, choroid plexus, neurogenic subventricular zone (SVZ), hippocampus anterior and posterior, hypothalamus, thalamus, caudate putamen, pons, medulla, cerebellum, olfactory bulb and three cortical regions, namely, motor, entorhinal and visual cortex (Fig. 1a). On average we found a read count of over 9.5 million per group, when aggregating all samples according to brain region and age (Supplementary Fig. 1a) and over 55% of reads mapped to miRNAs (Supplementary Fig. 1b). 828 of 844 samples (98%) passed quality control (Supplementary Data 1 and 2). A subsequent UMAP visualization of the features (miRNAs, lncRNAs, piRNAs, rRNAs, scaRNAs, snoRNAs, snRNAs, and tRNAs) showed a distinct separation of several brain regions, e.g. olfactory bulb and cerebellum (Supplementary Fig. 1c). Neither sex nor age was identified as strong drivers for grouping (Supplementary Fig. 1d and 1e). Analyzing the composition of expressed counts per brain region indicated a homogenous distribution (Supplementary Fig. 1f). Within most brain regions like cerebellum, motor cortex, hippocampus anterior and olfactory bulb, the composition of expressed counts for all RNA classes remained constant over all ages (Supplementary Fig. 1g). Correlating each feature with age per brain region revealed 720 positive correlated tRNAs and 127 negatively correlated miRNAs (51 positively correlated miRNAs, Fig. 1b). We exemplary highlight a tRNA (tRNA-Glu-TTC-1-1), which was significantly positively correlated in both sexes in multiple brain regions (Fig. 1c). Motivated by these high feature numbers exhibiting a correlation with age, we visualized tRNA UMAP results, which failed to provide a distinct separation into brain regions, sex or age (Fig. 1d, Supplementary Fig. 2a and 2b). A PVCA explained 36% of the observed variance with brain region identity (Fig. 1e). In contrast, miRNAs exhibit a larger share of variance (54%) explained by brain region. Also visible in the clear UMAP separation for the 1,174 miRNAs by brain region as driving factor over sex or age (Fig. 1f, g, Supplementary Fig. 2c–e). Age, as an independent factor, introduced 0.6% of the variation and 3.8% in combination with the brain region. The

variable sex was responsible for 0.2% of the variation and for 2.8% in combination with the brain region (Fig. 1g). Since our sequencing protocol is optimized for miRNAs and we found the greatest region-specificity and high age-correlation within miRNAs, we focused the following analysis on miRNAs. An overview of our miRNA expression data can be found at https://ccb-compute2.cs.uni-saarland.de/brainmirmap.

We herein present a study analyzing miRNA expression in a spatial resolution of 15 regions within the male and female brain using NGS. Previously, only micro-array-based studies were performed with up to 13 regions in male mice[18,19]. Using NGS methods combined with a greater sample size of each sex enabled us to identify more expressed miRNAs. Furthermore, with NGS isomiR alterations of miRNAs can be identified.

First, we focused on identifying strong region-specific signatures of miRNA expression in the different brain regions for both sexes, unified and individually. Therefore, we analyzed the expression of young adult mice (ages: 3, 12, and 15 months). Visualizing the UMAP results exhibited a clear separation driven by the brain region identity (Supplementary Fig. 2f). Exceptions like the overlap of hippocampus anterior and posterior can be explained by their anatomical and functional proximity. Neither sex nor age can be identified as strong factors driving a distinct grouping in the UMAP visualization (Supplementary Fig. 2g and 2h).

To identify miRNAs driving the region-specific grouping (Supplementary Fig. 2f), we selected 50 miRNAs according to the highest coefficient of variation (Supplementary Fig. 2i). We identified miRNAs which were distinctly expressed with respect to the brain average (highlighted with a black border) and clustered the respective brain regions into four clusters (cf. Methods). The strongest region-specific signature was observed in the olfactory bulb. Among the 15 distinctly expressed miRNAs were miR-200a-5p, miR-200a-3p, miR-200b-5p, miR-200b-3p, and miR-200c-5p. MiR-200 family members in the olfactory bulb are crucial to mediate neuronal maturation through targeting *Zeb2* during postnatal development[22].

Splitting the data into male and female samples, however, revealed differing signatures (Fig. 2a), underlining the importance of a sex-separable dataset. For the male samples, the strong unique signature of the olfactory bulb persisted. For the females, the signature is less pronounced, but a distinct expression miR-200 family members persisted (miR-200b/c-5p). In females, pons and medulla exhibited strong regional expression signatures, by each forming a separate cluster. MiR-10b-3p and miR-10b-5p were the signature miRNAs in the medulla and shared between medulla and pons were miR-1a-1-5p and miR-10a-3p/5p. MiR-1 and miR-10b were previously reported to regulate *BDNF* (brain-derived neurotrophic factor) an important protein involved in synaptogenesis, memory, and leaning[23]. In males, a strict anatomical organization of miRNA expression was observed as the olfactory bulb anterior had a distinct profile differing from the central brain regions (corpus callosum, subventricular zone, hypothalamus, thalamus, choroid plexus) and the posterior regions, medulla, pons, and cerebellum.

We determined sex-unspecific miRNA regional expression patterns (Fig. 2b, Supplementary Data 3): in medulla, we observed thirteen sex-unspecific miRNAs (e.g., miR-10a/b), eleven in the olfactory bulb (e.g., miR-200b/c-5p), nine in pons, miR-195b in cerebellum, and miR-653-5p in hypothalamus. In contrast, we identified sex-specific miRNAs with regional expression patterns in thirteen regions for males (e.g., miR-133b-5p in cerebellum) and in all regions for females (e.g., miR-471-3p in medulla). Hence, we concluded that regional miRNA expression patterns occur in a sex independent manner in anterior and posterior brain regions, and strong sex-specific regional patterns rather occur in central brain regions (corpus callosum, motor cortex, choroid plexus, thalamus).

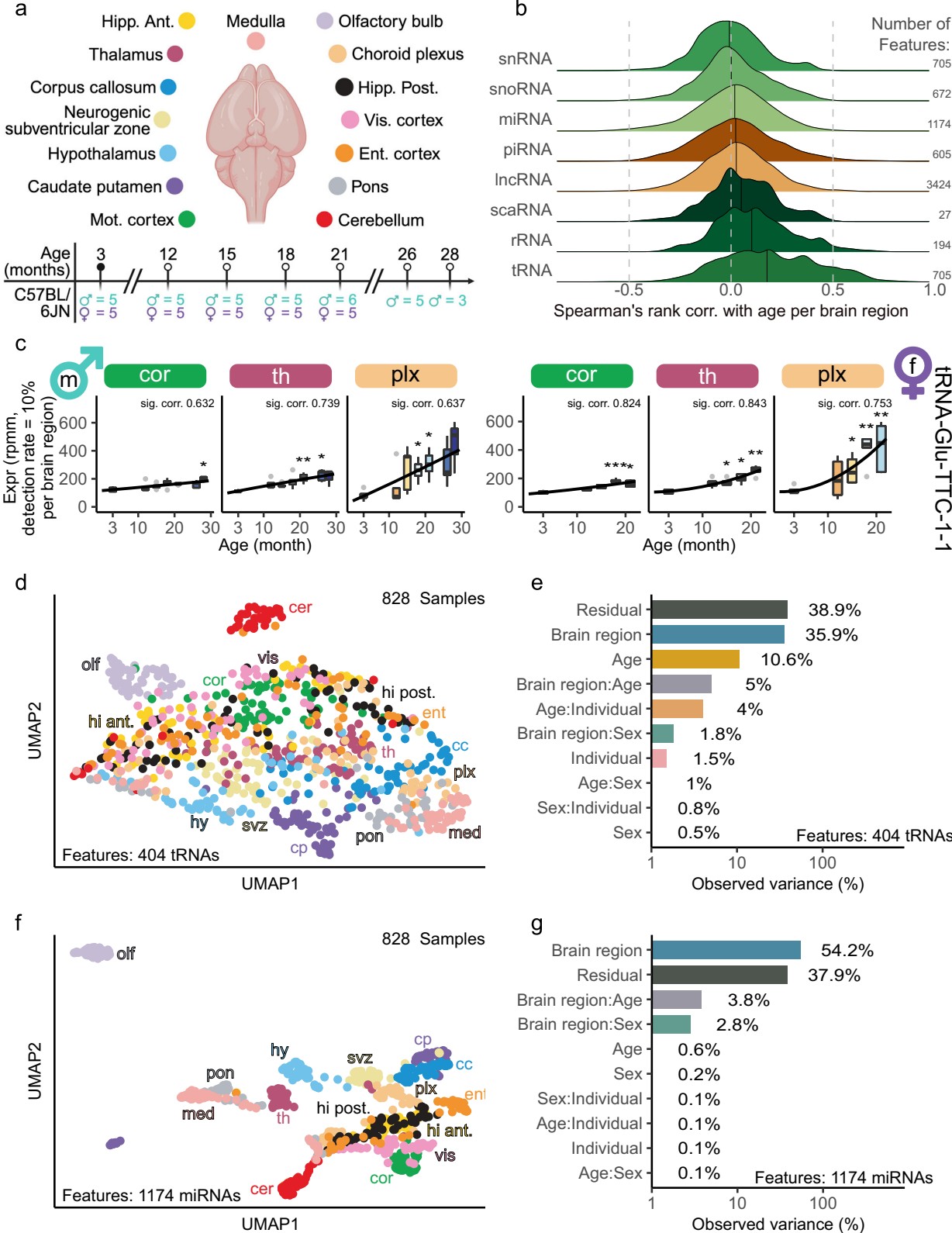

## Sex as a factor for miRNA expression during aging expression patterns

Considering the importance of sex as a variable in regional expression patterns, we chose to investigate the impact of sex as a variable within our dataset compared to age. During sex-specific analysis, we were limited to 14 brain regions and timepoints up to 21 months due to an insufficient number of older female samples.

Region-wise analysis of variation shares revealed that for two regions - mot. cortex and caudate putamen - the variation introduced by the variable sex was 20.2% and 20.3%, respectively (Fig. 2c). Whereas the variation share of age was under 12% for both regions. Sex differences in pathologies as well as neural properties and associated mechanisms have been reported previously for caudate putamen[24], e.g., in humans for fiber connection strength.

**Fig. 1 | Spatio-temporal overview and analysis of RNA expression in the aging mouse brain. a** Study overview: Brain tissues collected at 3, 12, 15, 18, 21, 26, and 28 months from 15 brain regions defined by the Allen Brain Atlas. Numbers below the timeline indicate mouse individuals per age and sex. Created in BioRender. Engel, A. (2025) https://biorender.com/w13g377. **b** Distribution of Spearman's rank correlation coefficients between age and features per RNA class calculated separately for each brain region using all samples. The black lines within the ridgeline indicate medians. The feature counts are displayed on the right. **c** Time series boxplots for tRNA-Glu-TTC-1-1 expression in three brain regions, separated by sex. Spearman's rank correlation coefficients from the tRNA expression with age are displayed above each plot (significant, if adjusted $p$-value < 0.05, two-sided Spearman's rank correlation test adjusted and adjustment for multiple testing using Benjamini-Hochberg procedure (cf. Methods)). Asterisks mark significant

deregulation (fold change $\geq 1.5$ or $\leq 1/1.5$ and adjusted $p$-value < 0.05 from two-sided Welch's $t$-test, Benjamini-Hochberg procedure) of the older ages to the reference age (3 months). The source data contains all exact values and sample sizes. Box borders correspond to the 25th ($Q_1$) and 75th Percentile ($Q_3$), the middle line to the median and whiskers to the minimum (maximum) of the minimum value or $Q_1 - 1.5 \cdot$ IQR (maximum value or the $Q_3 + 1.5 \cdot$ IQR) where IQR determines the interquartile range. Solid grey dots in the plot indicate the potential outliers in the data. **d** UMAP of all samples for tRNA features, colored by brain regions (Fig. 1a). **e** Principal Variance Component Analysis for tRNAs across all samples showing variance explained by brain region, age, sex, and individual. **f** UMAP of all samples for miRNA features colored by brain regions (Fig. 1a). **g** Principal Variance Component Analysis for the miRNAs across all samples, showing variance explained by brain region, age, sex, and individual. Colors as in Fig. 1e.

We identified 86 significantly upregulated and 37 significantly downregulated miRNAs between male and female in the caudate putamen (Fig. 2d, cf. Methods). In the motor cortex, 26 miRNAs were significantly upregulated and one miRNA significantly downregulated. Thalamus and corpus callosum also exhibited solely sex-driven variances above 10% but equally strong solely age-driven variances. In the thalamus, however, only seven miRNAs were significantly upregulated between male and female. In contrast, regions with a low share of variation explained by sex, such as olfactory bulb exhibited no significantly differentially expressed miRNAs. A gene set enrichment analysis (GSEA) over the miRNAs in mot. cortex, caudate putamen, and thalamus revealed overlaps between regulated pathways (Fig. 2e). E.g., "cerebral cortex radially oriented cell migration" was depleted in all three regions and "complement-dependent cytoxicity" was enriched. In summary, we found sex-specific miRNA expression patterns that persist during the entire lifespan and dominate over age-related expression changes, especially strong in the motor cortex and caudate putamen. These changes potentially relate to sex-specific regulation of distinct pathways.

However, our study focusses on age-related miRNA expression patterns and for five regions the share of variation explainable by age (>12%) dominated over the share explainable by sex (<10%) and the combination between age and sex (<3%) (Fig. 2c). These regions, namely, subventricular zone, olfactory bulb, vis. cortex and medulla and choroid plexus (18.5% of variation explainable by age) were especially interesting for the question whether age-related miRNA changes in the brain occur region specifically.

### Sex-specific miRNA expression changes during aging

We first analyzed male and female samples separately to be able to detect common and sex-specific effects. By collecting multiple discrete age stages, we were able to approximate the aging trajectories over the lifespan for each miRNA in each region and for both sexes. Analyzing trajectories is crucial as expression changes can occur in differing degrees and varying shapes, but in similar directions.

As miR-9 family members were previously extensively studied in the brain[25–27], we exemplarily investigate miR-9-5p expression. MiR-9-5p is highly expressed in all regions (Supplementary Fig. 2j), though it is not amongst the previously defined 50 most variable miRNAs (Supplementary Fig. 2i). In the olfactory bulb, we detected a median expression of over 100k rpmm. During aging, the miR-9-5p expression remained mostly stable in medulla and motor cortex in both sexes (Fig. 3a, Supplementary Fig. 3a). However, the expression in the olfactory bulb slightly decreased over the lifespan in males but remained mostly stable in females. MiR-9-5p is known to be involved in neurogenesis, axon development, differentiation, and proliferation of neural progenitor cells[27]. Hence, further investigation of functional consequences of its region-specific and sex-specific aging expression could yield to additional insights.

We measured the relation of each miRNA with age in each brain region by calculating the Spearman's rank correlation coefficient in males and females. Across all brain regions, we observed significant correlations with age for 13.79% of the miRNAs in males and 37.87% in females (Supplementary Fig. 3b). In males, 12 miRNAs were significantly positively correlated in the vis. cortex and in olfactory bulb 12 miRNAs were significantly anti-correlated (Fig. 3b, Supplementary Data 4). In females, 63 miRNAs were significantly positively correlated in choroid plexus and 13 were significantly anti-correlated in the SVZ. Three miRNAs in males and four miRNAs in females were significantly positively correlated with age in more than three brain regions (Fig. 3c). MiR-212-3p was significantly anti-correlated with age in males and miR-3473e in females, both for two brain regions, namely olfactory bulb and corpus callosum, hippocampus anterior and olfactory bulb, respectively. MiR-155-5p, miR-146a-5p, and miR-5100 were significantly positively correlated with age in males and females, exhibiting a strong aging signature independent from sex and region. Given that both miR-155-5p and miR-146a-5p regulate neuroinflammation and are implicated in neurodegenerative diseases[28], their roles in brain aging present compelling targets for further investigations.

By calculating correlations, we neglected non-monotonic effects in the trajectories. Therefore, we considered the differences between the older ages (12 to 28 months for male and 12 to 21 months for females) and 3 months by performing a differential expression (DE) analysis using these comparisons (Supplementary Data 5, cf. Methods). In the choroid plexus in males, exclusively one miRNA (mmu-miR-5100) was significantly positively correlated with age. However, more than 280 miRNAs were upregulated at 15 months and all later time points in males in this region (Fig. 3d). Out of these 280 miRNAs, 204 miRNAs were consistently upregulated in all consecutive time points from 15 to 28 months. These results indicate that the expression is drastically increased between 12 and 15 months and remained constantly high thereafter. In females, we observed 76 significantly positively age-correlated miRNAs (Fig. 3b) and a matching high count of upregulated miRNAs is observed within the age comparisons across all regions (Fig. 3d). Especially the brain regions cerebellum and choroid plexus stood out with 511 and 308 distinct miRNAs over all age comparisons, respectively. While the cerebellum exhibited an even higher upregulation trend in females starting at 12 months of age with 451 upregulated miRNAs, no significant age-correlated miRNAs were observed.

Though we observed strong deregulation trends for many miRNAs in different regions, few were significantly deregulated (Supplementary Fig. 3c). Therefore, we aggregated the miRNAs that were significantly deregulated uniquely and in multiple regions to determine the strongest age-deregulated miRNAs per sex (Supplementary Fig. 4a). MiR-5100, also identified as a significant age-correlated miRNA in multiple brain regions in both sexes (Fig. 3c), was significantly upregulated in males in the medulla and corpus callosum (Fig. 3e).

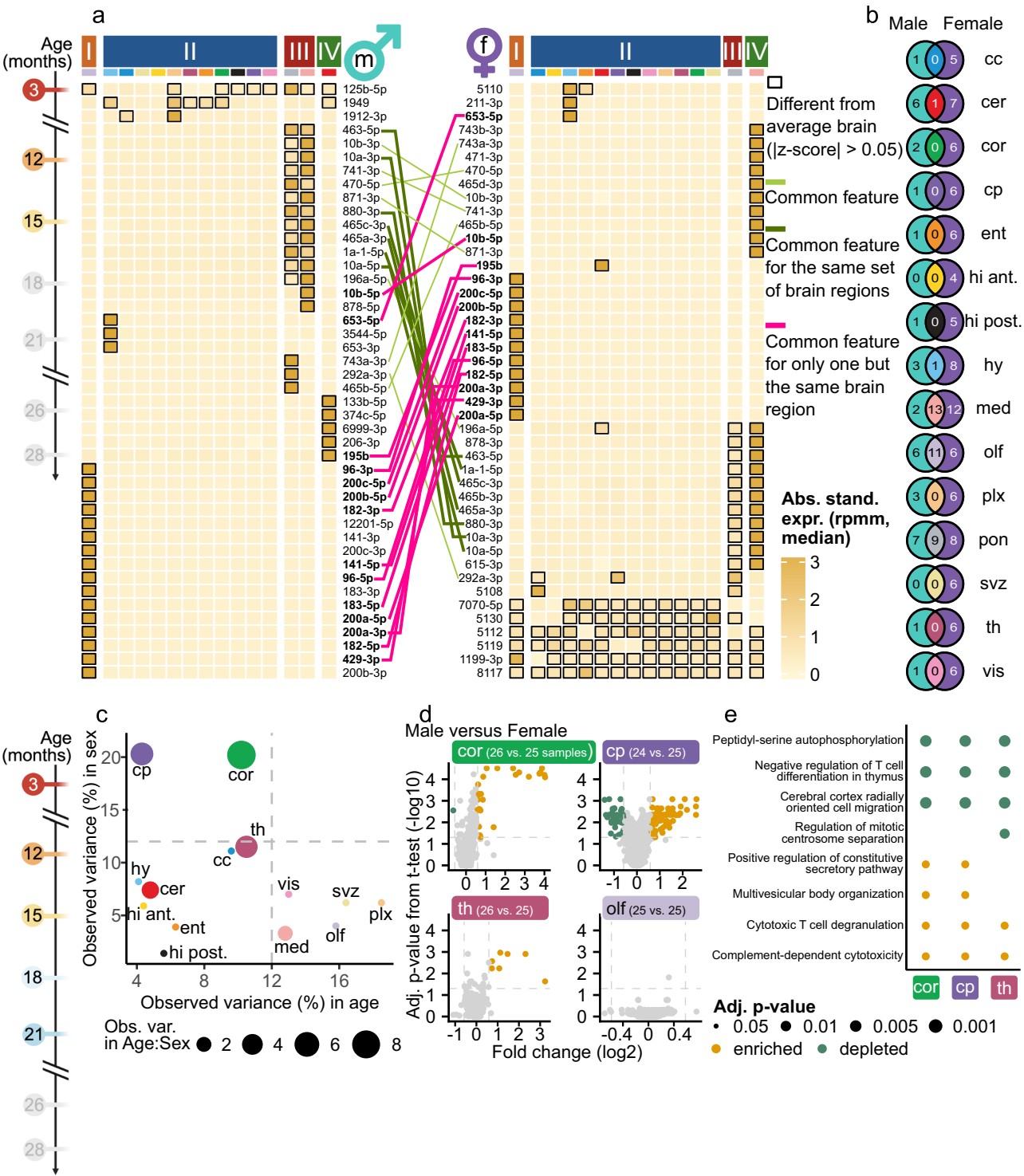

More miRNAs were significantly downregulated in unique regions in males than in females (Supplementary Fig. 4a).

We aggregated the miRNAs obtained from the age-correlation and deregulation analysis to a candidate set presenting the age-related miRNAs (Supplementary Fig. 4b, Supplementary Data 6). In males, corpus callosum, visual cortex, and olfactory bulb were the most prominent regions affected by miRNA expression changes, as opposed to choroid plexus in females. We summarized the region-specific and age-related signatures in each sex (Fig. 3f). Even though there is a high number of significant age-related miRNAs, we found no region-specific miRNA that also exhibited strong age-related expression changes in any region in males. In females, we detected region-specific miRNAs as

well as significant age-related ones, but again, no simultaneously region-specific and age-related miRNAs were detected. Our findings indicate that miRNAs exhibiting a strong regionally defined expression pattern in the brain have a stable expression during aging. In contrast, even though age-related miRNAs occur in a region-dependent manner, these miRNAs were not amongst the highest variable ones between all regions.

To see whether miRNA candidates from the sex-specific bulk mouse data showed similar trends in humans, we investigated miRNA expression within the ROSMAP dataset[29]. Between the oldest (92–102 years) and the youngest (71–80 years) group, we found 144 down-regulated and 10 upregulated miRNAs in females and 42

**Fig. 2 | Sex- and brain region-specific miRNA variation in the mouse brain.**
**a** Heatmaps of the 50 top miRNA from brain regions determined by the coefficient of variation calculated using the medians of the expression values of each brain region. Shown are the absolute standardized expression values (z-scores) for the younger male (left) and younger female (right) samples. Black borders indicate the binarization (|z-score|> 0.5) on which a clustering into four clusters using a hierarchical clustering was applied. We consider a miRNA in a brain region different from the average brain if it surpasses the aforementioned threshold. If it only exceeds the threshold for one brain region, we call it brain region-specific. Lines connecting the two heatmaps are highlighting the occurrence of common features (light green). If features are differing from the average brain for only one but the same brain region in both sexes, they are highlighted in pink and in dark green if they differ from the average brain in the same set of multiple brain regions. For

visualization purposes, we removed features entirely below the selected threshold. **b** Venn diagram linking the features which are differing from the average brain per brain region between the male and female heatmap. **c** PVCA showing the observed variance for each brain region individually over all sex-matched samples (3, 12, 15, 18, 21 months) for age and sex. The point size indicates the variance when observing age and sex in combination. Colors refer to the brain regions (Fig. 1a). Thresholds at 12% for both axes are marked with grey dashed lines. **d** Volcano plot for the sex-specific comparison of mot. cortex, choroid plexus, thalamus and olfactory bulb. Colored dots indicate significantly down (green) and significantly upregulated (yellow) miRNAs (fold change ≥ 1.5 or ≤ 1/1.5, adjusted *p*-value < 0.05, two-sided Welch's *t*-test, Benjamini-Hochberg procedure). **e** The gene set enrichment analysis (GSEA) results obtained from MIEAA[60] for mot. cortex, choroid plexus, thalamus showing the top 10 depleted (green) and enriched (yellow) pathways (cf. Methods).

downregulated and 32 upregulated miRNAs in males (Fig. 3g, cf. Methods). We considered the deregulated miRNAs with an absolute effect size greater than or equal to 0.5 as age-related candidates (11 downregulated and 4 upregulated in males; 19 downregulated in females, Fig. 3h). We intersected these with mouse age-related miRNAs in mot. cortex, the closest matching brain region to the human data (dorsolateral prefrontal cortex)[30]. We found no overlap of age-deregulated miRNAs in the same direction. As the youngest human age of 71 years roughly compares to the last time point of the mouse data (28 months), it is possible that the trends observed within the mouse data could be present in humans but cannot be captured during this short time course at the end of the human life span.

## Sex independent miRNA expression changes during aging

Apart from sex-specific miRNA expression changes, we recognized common signatures that we aim to verify in a joint analysis. Analyzing all samples together enables us to additionally investigate the pons. We calculated region-wise Spearman's rank correlation coefficients of miRNA expression with age (Supplementary Fig. 4c, Supplementary Data 7) and aggregated the results resolved in unique and multiple miRNAs (Supplementary Fig. 4d). We found less miRNAs significantly positively correlated with age in the combined analysis and more miRNAs significantly anti-correlated, especially driven by 75 anticorrelated miRNAs in pons. We found 11 miRNAs significantly anti-correlated (e.g., miR-18a-5p in olfactory bulb and pons) and 7 significantly positively correlated in multiple tissues (e.g., miR-155-5p in corpus callosum, mot. cortex, caudate putamen, medulla, choroid plexus, SVZ, thalamus, and vis. cortex). Hence, we were able to detect more age-correlated miRNAs in the combined analysis than in the sex-separated one due to the higher sample size.

Comparing deregulated miRNAs within each region and age, revealed a persisting aging signature in the combined analysis (Fig. 4a, Supplementary Data 8). The number of miRNAs downregulated in pons varied from 46 miRNAs (15 months) to 397 miRNAs (18 months) (Fig. 4a). MiR-9-3p and eleven other miRNAs were downregulated in each comparison between 12 months to 28 months. In hippocampus posterior 196 miRNAs were commonly downregulated at 26 and 28 months of age, even though no significantly anti-correlated miRNA was found. This indicates a strong expression decrease between 21 and 26 months, which was not detectable via correlation. In choroid plexus, we determined 45, 122, and 179 significantly upregulated miRNAs at 15 months, 18 months, and 21 months, respectively (Supplementary Fig. 4e). Seventeen of these were commonly deregulated in all three comparisons (including miR-155-5p and miR-146a-5p). An aggregation of the significant results resolved showed a high number of significantly upregulated miRNAs in choroid plexus (214 miRNAs) and significant downregulation in pons (94 miRNAs), SVZ (82 miRNAs) and choroid plexus (54 miRNAs) (Supplementary Fig. 4f). We observed deregulation for all brain regions and age comparisons to 3 months for all expression levels (Supplementary Fig. 4g). In our previous study, which examined the expression by bulk sequencing, we observed

peaks of deregulation at 12 and 18 months in the brain without any regional resolution[31]. The fact that we did not observe these peaks across all brain regions in this study highlights the importance of a region-specific investigation.

We compared miRNAs identified as age-related within each brain region, evaluating unique and overlapping miRNA sets analog to the sex-separated analysis (cf. Methods; Fig. 4b). There were many miRNAs uniquely associated with age in one region, especially 134 in the choroid plexus and 82 in pons. However, 41 common miRNAs were associated with aging in both these regions. The SVZ and choroid plexus share 31 common aging miRNAs. SVZ, choroid plexus, and pons shared 11 miRNAs, comprising the highest number of shared miRNAs and regions. MiR-5100 as a cross-region age-related miRNA was observed in ten brain regions (corpus callosum, cerebellum, ent. cortex, medulla, olfactory bulb, choroid plexus, pons, SVZ, thalamus and hippocampus anterior), miR-146a-5p in six (corpus callosum, mot. cortex, caudate putamen, olfactory bulb, choroid plexus and thalamus) and miR-155-5p in eight (corpus callosum, mot. cortex, caudate putamen, medulla, choroid plexus, SVZ, thalamus and vis. cortex). These observations indicated miR-155-5p, miR-146a-5p, and miR-5100 as cross-sex brain aging miRNAs (Supplementary Data 9). To investigate the functionality of the identified age-related miRNAs, we compared the candidates per brain region to the list of cholino-miRNAs[32]. Overlaps between these miRNAs regulating cholinergic genes and age-related miRNAs could indicate that the miRNA expression changes partially relate to altered acetylcholine signaling in aged individuals. We observed an overlap of seven age-related miRNA with the cholino-miRNAs in 7 brain regions (Supplementary Fig. 4h). In particular, mmu-miR-146a-5p, one of the cross-region age-related miRNAs is a known cholino-miRNA.

Further, we examined whether the miRNAs expression changes are driven by aging or can be related to transcription process alterations. Per brain region, we examined a +/−10kb window around each significantly age-correlated miRNA on the same and the opposite strand for occurrence of significantly age-correlated miRNAs (cf. Methods, Supplementary Fig. 4i). As a reference, the average number in this defined neighborhood is 5.27 on the same strand and 0.25 miRNAs on the opposite strand. No neighboring significant age-related miRNA was found on the opposite strand. Additionally, we visualize the significantly correlated miRNAs with respect to their cumulative genomic coordinates (Supplementary Fig. 5a). Our data indicate no significant enrichment on the same or the opposite strand. This suggests that the observed miRNA changes are likely not a result of strand-specific transcriptional effects.

We performed a GSEA over the age-related miRNAs in all brain regions to explore the targeted pathways. This analysis revealed overlaps between six brain regions for the 50 most significant enriched and depleted pathways (Supplementary Fig. 6a). "GABA-ergic synapse" is depleted in hypothalamus, "Positive regulation of acetylcholine secretion, neurotransmission" in hypothalamus and thalamus, and "Axon development" in pons. "Apoptotic process" is enriched in

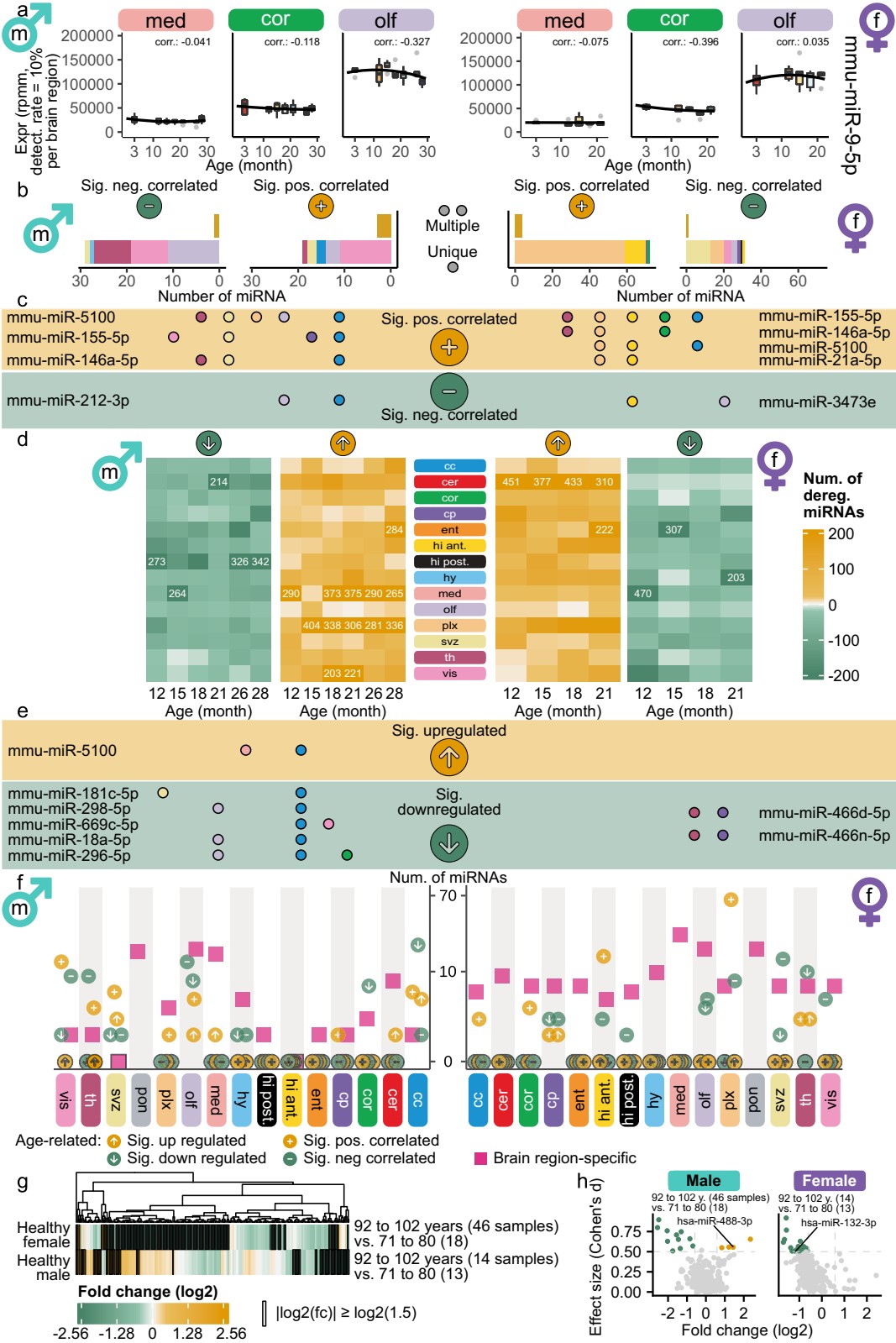

choroid plexus and vis. cortex and "DNA damage response" in choroid plexus.

**Region-specific cluster organization of aging miRNA trajectories**

To uncover continuous changes occurring over the entire lifespan other than correlations, we calculated miRNA aging trajectories for each miRNA in each brain region. This analysis enabled us to study the time-related changes in greater detail compared to studies using single young versus aged comparisons. Subsequently, we clustered these trajectories in 53 clusters to identify common trajectories (Supplementary Fig. 7a, Supplementary Data 10).

Twenty clusters exhibited a strong region-specific signature as more than 30% of the trajectories in the cluster originate from the same brain region (Fig. 4c). Cluster 7, 35 and 47 revealed three distinct

**Fig. 3 | Sex-specific miRNA dynamics in brain aging across male and female.** Sex-specific analysis, with male samples shown on the left and female on the right. **a** Time series boxplots of mmu-miR-9-5p in three brain regions. Spearman's rank correlation coefficient of miRNA expression with age is displayed above each plot (significant, if adjusted $p$-value < 0.05, two-sided Spearman's rank correlation test, Benjamini-Hochberg procedure). Boxplots follow Fig. 1c and Methods; exact values in Supplementary Data 4 and 5. **b** Barplots of significantly anti- or positively correlated miRNAs with age per brain region ($|R| \geq 0.5$, Spearman's rank correlation coefficient, adjusted $p$-value < 0.05, two-sided Spearman's rank correlation test, Benjamini-Hochberg). Upper bars contain miRNAs in the same direction significantly correlated in multiple brain regions (yellow). Lower bars contain miRNAs unique for one region (colors refer to brain regions (Fig. 1a). **c** Features significantly correlated in multiple brain regions. Dots indicate brain regions, background color yellow (green), if significantly positively correlated (anti-correlated). **d** Heatmaps showing up- or downregulated miRNAs per brain region between older ages and 3 months (fold change $\geq 1.5$ or $\leq 1/1.5$). Numbers are shown if more than 200 miRNAs. **e** Significantly upregulated miRNAs (yellow; fold change $\geq 1.5$, adjusted $p$-value < 0.05) and significantly downregulated (green; fold change $\leq 1/1.5$) in multiple brain regions. Colors analog to Fig. 3c. **f** Summary of brain region-specific and age-related features. Purple markers indicate brain region-specific features from Fig. 2a. Arrows pointing to the top in yellow (bottom in green) stand for significantly upregulated (downregulated) miRNAs and positive in yellow (negative in green) signs for significantly positively (anti-) correlated miRNAs. **g** Human ROSMAP[29] data: Healthy samples split by age of death (71-80, 81-91, 92-102 years). Color indicates the log2-fold changes between the youngest and oldest group per sex. Black borders mark if the fold change exceeds $|log2(1.5)|$. Asterisks indicate significance (two-sided Welch's $t$-test, Benjamini-Hochberg). **h** Scatterplot of Cohen's $d$ against log2-fold change, colored if thresholds for fold change ($|log2(fc)| \geq log2(1.5)$) and effect size ($|d| \geq 0.5$) are exceeded (upregulation yellow and downregulation green). Labeled miRNAs were age-correlated in mot. cortex (Supplementary Fig. 4c).

pons-specific aging miRNA trajectories dominated by a peak at 15 months and an overall decrease (Fig. 4d). In contrast, we observed an overall increasing expression for three choroid plexus region-specific clusters (cluster 2, 11 and 28). A loss of choroid plexus gene expression has been described in aging[33], which could be due to increasing miRNA expression. Especially, clusters 28 and 41 are dominated by a steady expression increase within all miRNA trajectories (Fig. 4e). Cluster 41 exhibited no prominent region-specificity. Therefore, we chose to investigate another characteristic within the clusters: the occurrence of a single miRNA trajectory originating from different regions (Fig. 4c). We found miR-5100 trajectories within the cluster 41 from five different brain regions (Fig. 4e). Cluster 28 contained miR-203 trajectories originating from three different regions (cerebellum, mot. cortex, choroid plexus). This miRNA negatively regulates NF-κβ signaling and microglia activation in neuronal injury[34,35]. It is an interesting target in neuroinflammation regulation and brain injury, in which miR-203 inhibitors have already been successfully tested[36]. Two clusters (24 and 38) exhibited a continuous decrease in expression. Neither of these clusters were region-specific. But cluster 38 contained trajectories from miR-466b-5p/466e-5p, 467d-5p and 469-families originating from four different regions, which were already identified as age-related.

We investigated the region-specificity of the clusters opposed to the occurrence of miRNA trajectories of the same miRNA originating from different regions in detail (Fig. 4f). If a cluster has a high region-specificity, there are no more than two to three miRNA trajectory occurrences in most cases. In contrast, clusters with multiple miRNA trajectory occurrences tend to have lower region-specificity. Out of the ten clusters harboring more than four times a single miRNA trajectory from different regions, only three were deemed region-specific. Especially in cluster 3, we observed that miR-3473a occurred ten times while the second most abundant mmu-miR-3473b occurred seven times.

The cross-sex brain aging miRNA, miR-5100 was found multiple times in cluster 41. Apart from miR-5100, miR-155-5p and miR-146a-5p were significantly correlated with age in males and females and are known as key modulators of immune response[37]. We therefore investigated the clustering patterns of the three cross-sex brain aging miRNAs. MiR-146a-5p trajectory center lines show an overall increase in age (Supplementary Fig. 8a). Motivated by the cluster analysis, we explored miR-146a-5p expression trajectories for all brain regions which show an overall increase for most brain regions (Supplementary Fig. 8b). Looking at cluster center lines containing trajectories from miR-5100, we observed an increasing tendence (Supplementary Fig. 8c). MiR-155-5p trajectories were also observed in different clusters sharing an increasing center line (Fig. 4g). The center lines of the clusters containing the miRNA trajectories for mot. cortex and cerebellum (28) increased towards 26 months. As our initial analysis did not reveal the age-relation of miR-155-5p in cerebellum, we investigated miR-155-5p expressions in all regions (Fig. 4h, Supplementary Fig. 8d). We observed an increase of miR-155-5p to different extents in all regions.

## Microglial age-driven miRNA expression changes

The results of the miRNA trajectory clustering confirmed the three increasing cross-sex brain aging miRNAs. This prompted us to investigate whether there is a cell type driving their expression. Mapping bulk miRNA expression back to cell types is challenging as there is no robust high-throughput single-cell detection method for miRNAs. However, miRNA expression patterns were previously measured using Cre recombinase-dependent miRNA tagging[38]. MiR-155-5p and miR-146a-5p expression is likely driven by microglia, as both showed an over 10-fold enrichment in microglia compared to brainstem[38,39] (Fig. 5a, Supplementary Fig. 9a). Unfortunately, miR-5100 was not detected. To decipher whether the observed miRNA increase is driven by changes in cell type ratios or microglial expression changes, we sequenced microglia from young (3 months) and aged (21 months) mice collected via FACS (Supplementary Fig. 9b, 9c, Supplementary Data 1).

We compared the miRNA expression of our young/aged microglia to data from Walsh et al.[40] and found a strong overlap of expressed core miRNAs (Fig. 5b). The highest expressed miRNAs according to the median per samples were observed for young and aged, e.g. miR-181a-5p (Fig. 5c). Based on the coefficient of variation, we selected the top 25 miRNAs for clustering. We found few miRNAs with rather distinct expression patterns in individual samples (Fig. 5d). Hence, we checked for our three brain aging miRNAs specifically. MiR-155-5p and miR-146a-5p were expressed in microglia above a threshold of 3 rpmm, miR-5100 was not. Only miR-155-5p showed an increase in aged microglia (Fig. 5e). A detailed analysis of deregulated miRNA between these two groups revealed 59 miRNAs upregulated in aged microglia and 70 miRNAs downregulated (cf. Methods, Fig. 5f, Supplementary Data 11). Within the upregulated miRNAs, miR-155-5p exhibited the highest fold change (fold change: 3.58, raw p-value: 0.002 and adjusted p-value: 0.659). Hence, the increasing expression of miR-155-5p in several brain regions is likely driven by an increased microglial expression.

## MiR-155-5p as sex independent cross-region aging miRNA

Summarizing all analysis, we considered miR-155-5p the most promising candidate for investigation into its regulatory mechanisms. We examined whether its regulatory mechanisms contribute to known aging interventions such as dietary restrictions as well as young plasma injections, which were reported as beneficial for aged individuals[41,42]. Hence, we investigated the expression changes of miR-155-5p in acute dietary restriction (aDR) and after young mouse plasma injections

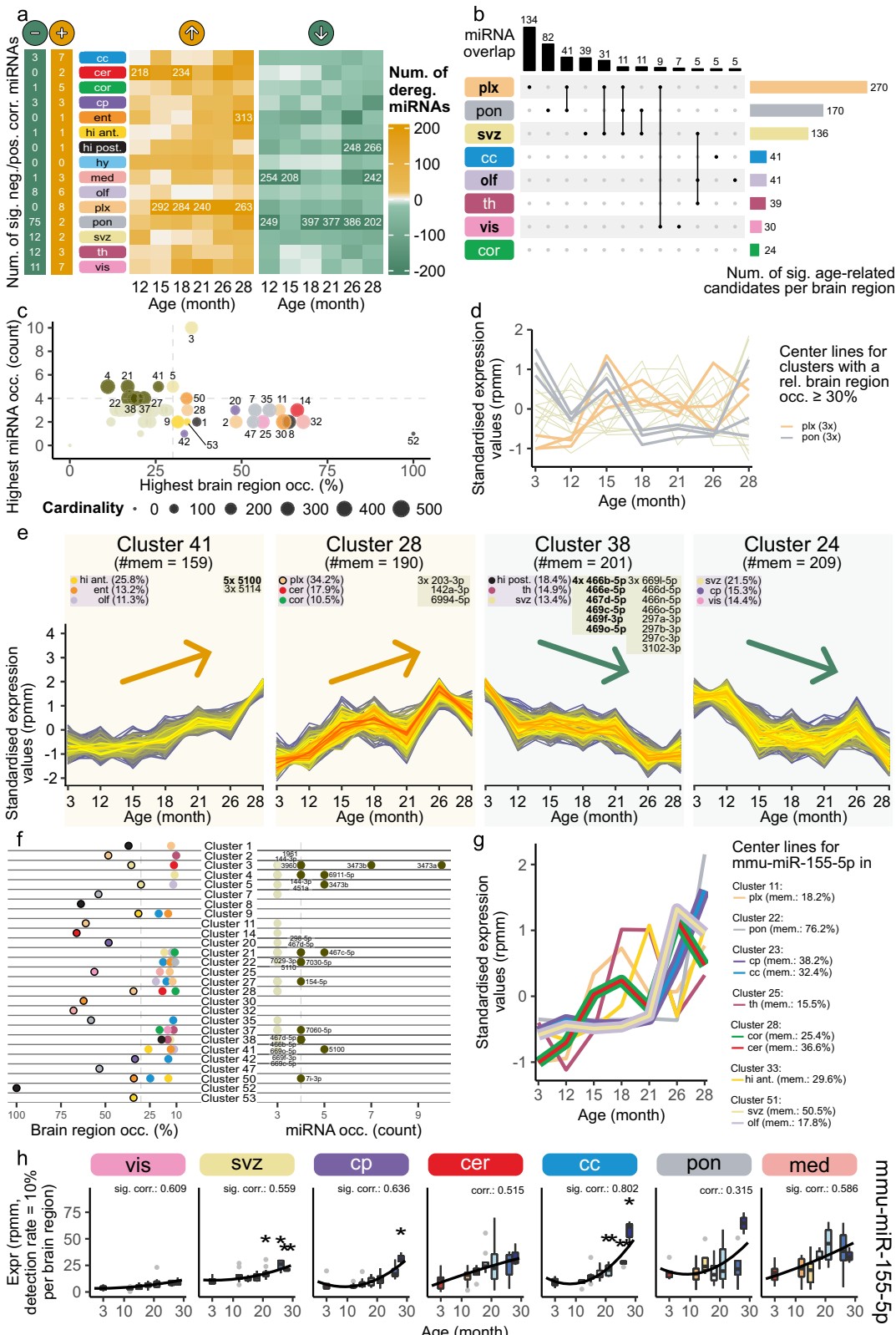

(YMP). We analyzed miRNA expression patterns in all previously collected regions of female mice aged 19 months, either fed *ad libitum* or treated with 4 weeks of aDR. And we analyzed samples from the young plasma injection cohort, which consisted of male 18-month-old C57BL/6JN mice injected with young mouse plasma[42]. We found an average read count per sample of over 21 million for aDR and 23 million for YMP (Supplementary Fig. 9d and 9e). On average, over 55% of the reads

were mapped to miRNAs (Supplementary Fig. 9f and 9g) and 113 of 118 samples (96%) for aDR and 68 of 84 (81%) for YMP passed our quality control (Supplementary Data 1). Across all regions, exclusively miR-451a showed a significant expression change during aDR (Supplementary Fig. 9h). An in-depth analysis of miR-155-5p expression in aDR did not reveal any persisting trends (Supplementary Fig. 9i). No significant expression changes were observed in any region after YMP

**Fig. 4 | Global miRNA expression dynamics in brain aging across regions.** All samples from the dataset are considered. **a** Heatmaps showing deregulated miRNAs between each older age and 3 months (fold change ≥ 1.5 or ≤ 1/1.5). Numbers shown if ≥ 200 miRNAs. On the left, significantly anti- or positively correlated miRNAs with age are given (|R| ≥ 0.5, Spearman's rank correlation coefficient, adjusted p-value < 0.05, two-sided Spearman's rank correlation test, Benjamini-Hochberg). **b** Upset plot of miRNAs changing significant with age per brain region, indicating unique and overlapping candidates. Significant candidates from Fig. 4a and Supplementary Fig. 4e. Brain regions above the age variance threshold in Fig. 2c are bold. Regions with <10 candidates and combinations with <5 miRNAs are omitted. **c** Trajectory clustering overview (k = 53, minimum membership 15%). Each dot represents a cluster, sized by cluster size. Dot color indicates brain region if occurrence exceeds 30%, dark green if the most frequent features incidence exceeds 4, otherwise light green. **d** Center lines of clusters with brain region occurrence >30%, highlighting choroid plexus and pons. **e** Trajectories of four selected clusters (z-scored miRNA expression). Yellow plots show a steady increase, green plots steady decrease with age. Labels show dominant regions (>10%) and frequent miRNAs (>3 features). Regions with black borders and bold miRNAs exceed the thresholds from Fig. 4c. **f** Detailed view of brain region occurrence (left) and miRNA frequency (right). Brain regions (<10%) and miRNA (<3) are omitted. Dots over thresholds from Fig. 4c (grey dashed lines) appear in dark green for miRNAs or have black borders for brain regions. **g** Center lines of seven clusters containing mmu-miR-155-5p trajectories. Line widths vary if multiple regions share a center line. **h** Boxplots of mmu-miR-155-5p trajectories in seven brain regions. Asterisks highlight significant comparisons to 3 months (fold change ≥ 1.5 or ≤ 1/1.5, adjusted p-value < 0.05, two-sided Welch's t-test, Benjamini-Hochberg procedure). Spearman's rank correlation coefficient with age are displayed above each plot (significantly, if adjusted p-value < 0.05, two-sided Spearman's rank correlation test, Benjamini-Hochberg). Boxplots follow Fig. 1c and Methods; exact values in Supplementary Data 7 and 8.

(Supplementary Fig. 9j). Similarly, for YMP, we did not detect any significant miR-155-5p expression changes compared to control groups, however we observed decreased expression in corpus callosum, cerebellum, caudate putamen, medulla, and choroid plexus and an increase in mot. cortex (Supplementary Fig. 9k). We hypothesize that the beneficial effects of these interventions are not crucially mediated via miRNA changes.

Sequence variations of mature miRNAs, so called isomiRs, occur naturally due to either altered cleaving patterns or sequence editing. IsomiR patterns can vary between different organs and in disease to their archetype miRNA expression. IsomiRs can alter the target spectrum of a miRNA, hence we investigated whether the isomiR-archetype ratio is altered during the aging process. We found that isomiRs of miR-155-5p followed a similar trend of increase during aging in all regions, but the archetype form dominated (Fig. 5g, Supplementary Fig. 10a). Furthermore, we checked whether there was a region-specific signature of isomiR expression for miR-155-5p. We discovered that in medulla miR-155-5p isomiRs showed a distinct expression pattern (Fig. 5h).

As miR-155-5p increased during aging, we investigated whether this increase leads to altered regulation of its mRNA targets. Therefore, we gathered all miR-155-5p targets using miRTargetLink 2.0[43]. Leveraging the matching bulk mRNA data[3], we calculated the Spearman's rank correlation coefficients between miR-155-5p and each target within each region (cf. Methods). Twenty-six targets were significantly anti-correlated during aging in multiple regions (Fig. 5i).

Cyclin H (CCNH) and the basic helix-loop protein (ARNLT) were significantly anti-correlated in five and six different regions, respectively. The correlation strength in each region differs (Fig. 5j). In caudate putamen the miR-155-5p increase lead to a strong ARNTL decrease, whereas in medulla the decrease was less pronounced. We observed similar region-dependent regulation strengths for ZFP322A (zinc finger protein) and PTPRJ (protein tyrosine phosphatase). PTPRJ activates MAPK1, which is an important player during extracellular signaling and various cellular processes. Additionally, we considered the functionally validated targets genes[44]. Six of the 80 target genes showed a significant anti-correlation with mmu-miR-155-5p for at least one brain region, including the previously shown ARNTL (ADAM23, PCSK5, REPS2, NRCAM and NSG2, Supplementary Fig. 10b). We found multiple targets within important age-related pathways, such as regulation of cell communication and the mTOR signaling pathway (Fig. 5i). Consequently, we focused on the additional transcripts of the mTOR pathway[45] and their correlations with miR-155-5p expression (Supplementary Fig. 10c). For corpus callosum, caudate putamen, ent. cortex, hippocampus anterior, choroid plexus, SVZ and thalamus a significantly anti-correlation can be observed for multiple genes like PDPK1, YWHAZ and CYCS. This highlights that miR-155-5p deregulation might not only affect its direct targets but also other downstream

targets within this pathway. Using the data from Keele et al.[46], we explored the protein levels between young and aged mice in hippocampus, as we predicted that miR-155-5p expression increase, leads to gene silencing by targeting MEF2A in hippocampus. In males, MEF2A protein was less expressed in 18 months aged mice as compared to 8 months old mice (Supplementary Fig. 10d). We conclude that age-related microglial miR-155-5p expression changes likely regulate mTOR pathway gene expression.

## Discussion

In our aging brain region-specific miRNA atlas, we identified region-specific miRNA signatures, especially in the olfactory bulb and cerebellum in males and medulla and pons in females. We reproduced previous findings of miR-200 exclusively expressed in olfactory bulb in males and extend this knowledge to equal region-specific expression in females. This miRNA family plays a crucial role in neurogenesis previously[22]. Furthermore, we found strong female region-specific patterns in medulla and pons that did not overlap with male expression patterns. The identified region-specific miRNAs are therefore interesting targets to study sex-dependent region-specific functionalities.

In the motor cortex and caudate putamen, we determined a strong sex-driven expression signature persistent over the entire lifespan. In sex-separated analysis, we found the most pronounced miRNA expression changes in females in choroid plexus, SVZ, and hippocampus. While in males, expression changes in the visual cortex, olfactory bulb, and corpus callosum dominated.

The analysis of miRNA patterns in brain regions during aging revealed sex-independent miRNAs increasing with age in multiple brain regions, namely miR-146a, miR-155-5p, and miR-5100. Especially miR-155-5p, which we identified as a global aging miRNA before[31], is an interesting target for further studies. This miRNA has been shown to regulate leukocyte adhesion at the inflamed BBB[47] and has been associated with neurological disorders such as AD[10] and the murine model of multiple sclerosis[48]. We detected an increase of miR-155-5p expression in aged microglia. Increased secretion of miR-155-5p from microglia mediates inflammatory neuronal cell death and therefore plays a pro-inflammatory role[49]. Whether the increased microglial expression of miR-155-5p in age is limited to microglia or is actively secreted into the microenvironment remains to be studied. Secretion of miR-155-5p via e.g. exosomes could contribute to the inflammaging phenotype, as miR-155 has been proposed as a central regulator in CNS-related inflammation[50]. In the disease context (AD), miR-155 together with interferon-γ signaling mediates a protective microglial state[10]. A microglia specific deletion of miR-155 reduced amyloid-β pathology in AD mouse models but caused hyperexcitability and seizures[51]. The distinct isomiR expression in certain brain regions, like medulla calls for further investigation. In our dataset, we discovered

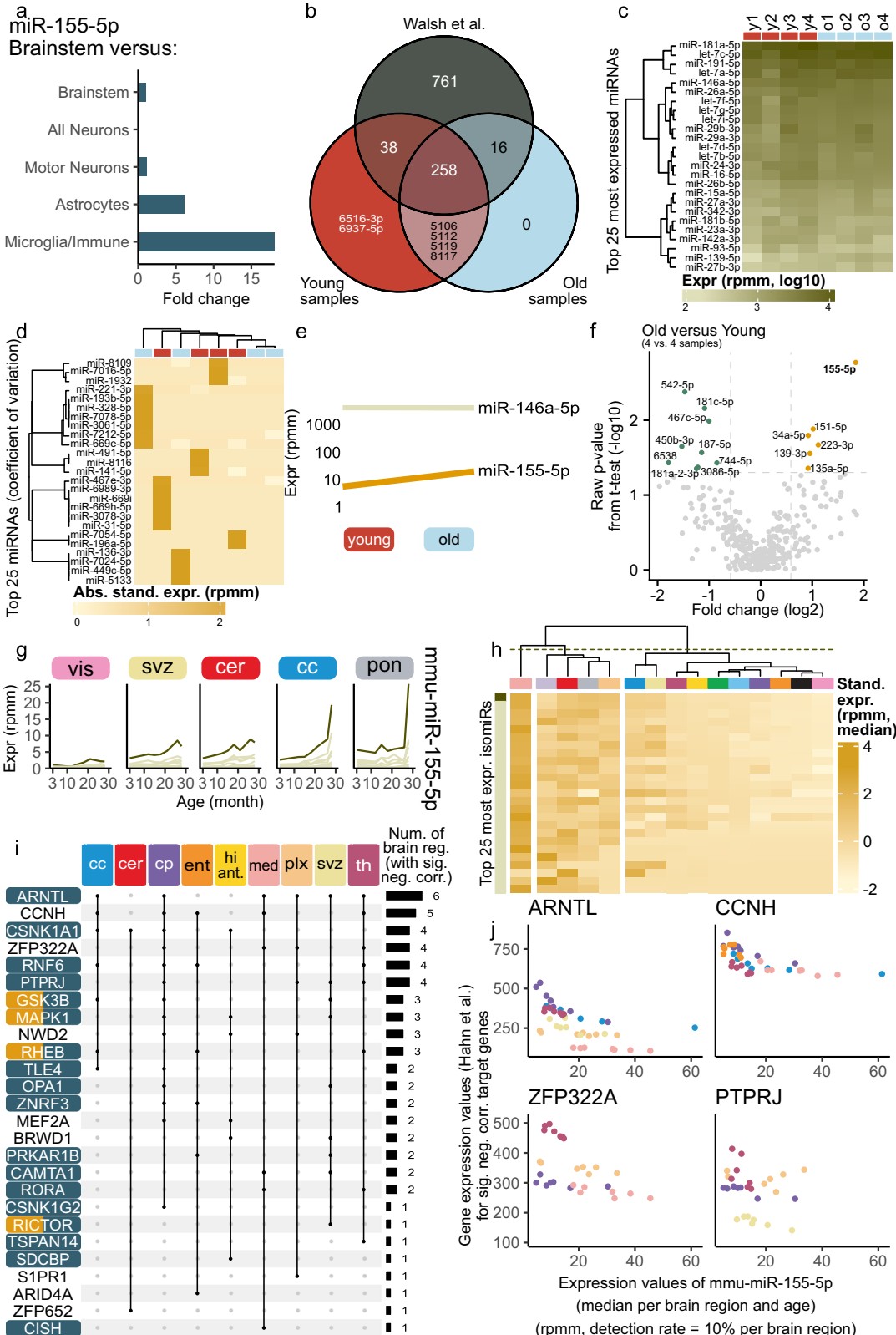

altered isomiR expression for miR-155-5p and an expression change in a target gene (*CISH*) exclusively in medulla. Additional studies could use this dataset together with functional validation experiments to advance understanding of isomiR regulatory properties, specifically in the aging context.

Limitations of our study include, aside from restricted transferability to humans, the limitations arising from technical challenges of

analysis of cell-type specific miRNA patterns. Commonly used single-cell sequencing techniques cannot be applied to assess mature miRNA expression due to the lack of polyA-tails, which are essential to most protocols. Hence, we resorted to FACS to sort for microglia. Selecting antibodies for the sorting process requires a careful balance to specifically select cells of interest. We chose to exclude CD45+ cells to distinguish resident microglia from infiltrating immune cells. However,

**Fig. 5 | MiR-155-5p expression and target interactions in brain aging and microglia. a** Based-on CNS microRNA Profiles[39]. Barplot displays fold changes of mmu-miR-155-5p between brain cell types and brainstem data. **b** Venn plot (distinct mode) of our microglia expression data (FACS-sorted young and aged mice) compared to immunopanning-derived profiles from young microglia[40]. **c** Top 25 most expressed miRNAs in our microglia dataset (expression in log10-scale), miRNAs clustered hierarchically. **d** Top 25 miRNAs by coefficient of variation, showing absolute standardized expression per sample and miRNA, clustered hierarchically. **e** Line plot for two miRNAs showing the median expression in young and old mice, indicating direction of deregulation. **f** Volcano plot of differential expression (old versus young), showing raw p-values (two-sided Welch's *t*-test). No miRNA reaches significance after adjustment, but mmu-miR-155-5p shows strong upregulation (marked in bold). **g** Trajectories of median expression per age for isomiRs of mmu-miR-155 in five brain regions. The canonical isomiR is highlighted as a dark line. **h** Top 25 most expressed isomiRs of mmu-miR-155-5p (sorted from top to bottom). Averaged standardized expression per brain region and isomiR, clustered by brain regions (hierarchical clustering with complete linkage) and split into three clusters. **i** Region-wise Spearman's rank correlation coefficient for target genes of mmu-miR-155-5p (from miRTargetLink 2.0[43]) using mRNA data from Hahn et al. [3]. Upset plot shows significantly anti-correlated genes (($|R| \geq 0.3$, adjusted *p*-value < 0.05, two-sided Spearman's rank correlation test, Benjamini-Hochberg procedure) with brain region assignment. Genes are colored by pathways: "mTOR Signaling Pathway" (yellow) and the "Regulation of cell communication" (blue). **j** Scatter plots for four of the significantly anti-correlated target genes (from Fig. 5i). Brain regions with significant anti-correlation are color-coded. Plots show the relationship between gene expression and the miR-155-5p, based on median values per brain region and age.

we thereby exclude a subset of resident microglia, as the expression of CD45 can increase with age in resident microglia. Another limitation is the question of whether different microglial states in aDR and YMP exist with respectively specific miRNA expression patterns. Changes in cell type composition and especially cell type states could drastically influence miRNA bulk expression patterns. These limitations further underline the need for studies of cell-type-specific miRNA expression and even different cell-type states. Due to limited access to aged mice, especially females, this was not possible in this study. Therefore, also our microglia data is also limited to male samples. In addition to human data, validation of brain aging miRNAs in other mouse strains would strengthen our hypotheses, as differing miRNA expression in different brain regions was previously reported[52].

In conclusion, we identified microglial-derived miR-155-5p as the most interesting therapeutic target. As it regulates broadly cellular communication pathways during aging and increased sex-independent manner across all brain regions. MiR-146a-5p is another interesting candidate, because we also observed its sex-independent cross-region expression increase. We furthermore identified unique aging signatures in e.g., choroid plexus, subventricular zone, and pons as interesting targets to study region-specifically. The sample size of this study enabled us to even identify sex-specific aging signatures that are currently heavily understudied. In sum, this atlas offers a comprehensive map of miRNA expression across different brain regions in a sex-specific manner, which was previously nonexistent, with possibility to go in-depth into isomiR analysis, including multiple time points to study aging signatures.

## Methods
### Samples
As previously described, male and female C57BL/6JN mice from the National Institute of Aging colony (Charles River) were shipped to the Stanford ChEM-H animal facility (Palo Alto), where they were housed for at least one month before euthanasia[3]. For each age group of 3, 12, 15, 18, and 21 months, 5 female mice and 5–6 male mice were used; age groups 26 and 28 months consisted only of 5 and 3 male mice, respectively. Animals were housed in cages of 2–3 mice, with a 12 h/12 h light/dark cycle, at 19.4–22.8 °C under 40–60% humidity and provided with food and water ad libitum. Over the course of four days, the sample collection was performed between 10 am and 12 pm. Mice were anaesthetized with 2.5% v/v Avertin, 700 µl of blood was drawn via cardiac puncture and followed by transcardial perfusion with 20 ml cold PBS. After immediate removal of the brains, the organs were snap-frozen by submission in liquid nitrogen-cooled isopentane (60 s) and ultimately stored at −80 °C before further processing. The respective regions were dissected via slicing and atlas-guided tissue punching while frozen. Using a metal brain matrix coronal sections of 1 mm thickness were sliced with 0.22 razor blades (Ted Pella, 15045; VWR, 55411-050). Regions of interest (1.5 mm and 2 mm diameter) were dissected quickly from the right

hemisphere of these sections using disposable biopsy punches (Alimed, 98PUN6-2, 98PUN6-3). The following 15 regions were collected: three cortical regions (motor cortex, visual cortex and entorhinal cortex), anterior (dorsal) and posterior (ventral) hippocampus, hypothalamus, thalamus, caudate putamen (part of the striatum), pons, medulla, cerebellum and the olfactory bulb, corpus callosum, choroid plexus and the subventricular zone. Four regions were collected in the following order, as the collection required overlapping punches: (1) motor cortex, (2) caudate putamen, (3) subventricular zone, (4) corpus callosum. All animal care and procedures complied with the Animal Welfare Act and were in accordance with institutional guidelines and approved by the institutional administrative panel of laboratory animal care at Stanford University. RNA was isolated using the RNeasy 96 kit (Qiagen, 74181) and a TissueLyser II (Qiagen, 85300), according to RNeasy 96 Handbook protocol "Purification of Total RNA from Animal Tissues using Spin Technology" without the optional on-plate DNase digestion.

### Aging interventions
Young Mouse Plasma (YMP) was collected following the protocol described by Villeda et al.[42] Briefly, C57Bl/6J male mice aged 2 months were anesthetized with 2.5% v/v Avertin after being group-housed. Around 700 µl of blood was drawn via cardiac puncture prior to transcardial perfusion. 15 µl of 250 mM EDTA (Thermo Fisher Scientific, 15575020) was used to collect blood. The mixture was centrifuged at 4 °C for 15 min at 1000 g to obtain plasma. The plasma from 20–25 mice was pooled together and dialyzed in 1X PBS using cassettes (Slide-A-Lyzer Dialysis Cassettes, 3.5 kDa molecular weight cut-off, 3–12 ml) before being frozen at −80 °C. For plasma transfer experiments, 18-month-old male C57BL/6JN mice were injected retro-orbitally with 150 µl of YMP per injection. Prior to injection, mice were habituated by being placed on the procedure table in their cage. Injections were administered every 3–4 days, alternating between the left and right eye to allow for recovery. Mice were rested for four days before tissue collection.

For the aDR study with C57BL/6JN mice, 18-months-old mice were randomly assigned to AL or aDR. aDR treatment was initiated by transferring mice from AL to 10% aDR for 7 days. After that, aDR was increased to 25%. aDR animals were fed once per day between 3–5 p.m., and all animals were checked daily for their well-being and any deaths. For the first 16 days, weights were checked daily. Mice were euthanized at the age of 19 months. All mice were euthanized in the morning within a period of 6 h prior to the regular feeding time of the DR mice.

The aDR study with C3B6F1 mice was performed in accordance with the recommendations and guidelines of the Federation of the European Laboratory Animal Science Association (FELASA), with all protocols approved by the Landesamt für Natur, Umwelt und Verbraucherschutz, Nordrhein-Westfalen, Germany (84-02.04.2015.A437). Female F1 hybrid mice (C3B6F1) were generated in-house by crossing

C3H/HeOuJ females with C57BL/6NCrl males (strain codes 626 and 027, respectively, Charles River Laboratories). Five female animals were housed as a group in individually ventilated cages under specific-pathogen-free conditions with constant temperature (21 °C), 50–60% humidity, and a 12 h/12 h light/dark cycle. For environmental enrichment, mice had constant access to nesting material and chew sticks. All mice received commercially available rodent chow (ssniff R/M-Low phytoestrogen, ssniff Spezialdiäten, Germany) and were provided with filtered water ad libitum. aDR animals received 60% of the food amount consumed by AL animals. aDR treatment was initiated at 20 months of age by directly transferring mice from AL to 40% DR. aDR animals were fed once per day, and all animals were checked daily for their well-being and any deaths. Mice were euthanized at the age of 24 months. All mice were euthanized in the morning within a period of 3 h prior to the regular feeding time of the DR mice. Mice were euthanized by cervical dislocation, and tissues were rapidly collected and snap-frozen in liquid nitrogen.

The cohort of mice treated with YMP or PBS was housed at the Palo Alto VA animal facility under a 12 h/12 h light/dark cycle at 68–73 °F under 40–60% humidity. All experiments were performed in accordance with institutional guidelines approved by the VA Palo Alto Committee on Animal Research. Euthanasia and organ collection were conducted in the same way as the aging cohorts.

## Microglia isolation

Microglia from young and aged mice (3 and 21 months, C57BL/6, males) were isolated via FACS-sorting following the protocol described below. In brief, mice were anaesthetized with Avertin and perfused with 20 mL ice-cold DPBS. Brains were dissected, hemispheres separated, and the olfactory bulb and cerebellum removed. Single-cell solutions were created for each hemisphere via mincing and douncing the tissue, the solution was filtered (70 μm cell strainers, Falcon 352350) and finally centrifuged (400 x g, 10 mins, 4 °C). After resuspension in MACS buffer and addition of myelin removal beads (Miltenyi Biotech, 130-096-433), solutions for each hemisphere were loaded on LD columns (Miltenyi Biotech, 130-042-901) to remove myelin. Columns were washed twice with MACS buffer. Cells were pelleted via centrifugation (400 x g, 5 mins, 4 °C) and resuspended in FACS buffer. FC blocking antibody was added and incubated for 5 mins. Primary antibodies (CD11b-FITC (1:50, Biolegend 101206); CD-45-BUV396 (1:50, Biolegend 50-162-785) and CD206-APC (Biolegend 141707)) were added and incubated for 30 min on ice. After another centrifugation, samples were resuspended in 0.5 mL FACS buffer, and Sytox Blue (Thermo, S34857) was added for live cell labeling. Cells were sorted on a MA900 Multi-Application Cell Sorter (Sony Biotechnology) and selected by gating for live single cells (FSC-A/SSC-A). Further, cells were gated for CD11b$^+$; CD45$^{low}$; CD206$^-$ and sorted directly into 1.5 mL tubes containing 50 μl FACS buffer. As CD11b$^+$/CD45$^+$ brain cells are mainly infiltrating macrophages[53], we exclude these macrophages with our sorting strategy. FACS-isolated CD11b$^+$/CD45$^{low}$/CD206$^-$ brain myeloid cells are referred to as microglia in this manuscript. Between 10,000 and 40,000 cells per sample were collected. For the aged mouse sample 4, two mice were pooled together to reach the threshold of minimal 10,000 cells as input for RNA isolation. After collection was completed 1 mL QIAzol was added instantly. After vortexing, samples were stored at −80 °C until RNA isolation. Since cell sorting provides little input material for RNA isolation, the standard miRNeasy microKit (QIAGEN, cat. no. 217084) was used with standard protocol for elution of separated fractions for RNAs above and below 200 nt, to concentrate the miRNA input for library preparation. Library preparation of microglia miRNAs was optimized and performed as described below, except for the RT-primer input, which was altered to a 1:5 dilution, and amplification PCR was 25 cycles long.

## Library preparation

The MGIEasy Small RNA Library Prep Kit (Item 940-000196-00) was used for library preparation on the high-throughput MGI SP-960 sample prep system according to the manufacturer's protocol. In principle, this library preparation method works by ligating 3'- and 5'-adapters to all RNAs in each sample. During reverse transcription (RT), specific RT primers that bind to the adapters are used to generate cDNA and introduce sample-specific barcodes. Amplification of this cDNA is performed via a 21-cycled PCR. Size-selection of this PCR product is performed via magnetic beads (AMPure Beads XP, Beckman Coulter). To focus on the small RNAs a size of around 110 bp was selected, this was checked using an Agilent DNA 1000 Kit (Agilent Technologies). The concentration of each sample was measured by a QuBit 1x dsDNA High Sensitivity Assay (Thermo Fisher Scientific). Each library in this diet study consisted of 16 samples, barcoded with the following barcodes: 1–4, 13–16, and 25–32. All samples of one library were pooled after concentration measurement in an equimolar fashion to reach a concentration of 4.56 ng μl-1 for each sample in each pooled library. After circularization, the pooled libraries were sent for sequencing.

## Sequencing & data analysis

Samples were single-end sequenced on the BGISEQ500RS using the High-throughput Sequencing Set (SE50) (Small RNA) as a service provided by BGI, Hong Kong. We utilized miRMaster 2.0[54] with standard settings on the given datasets which performed an alignment against the mouse genome (GRCm38) and a mapping against miRNAs using miRBase[55] (version 22.1) using Bowtie[56] (version 1.2.3) with the options "-m 100 −best −strata" to obtain the raw counts for miRNAs and their isomiRs, for lncRNAs, piRNAs, rRNAs, scaRNAs, snoRNAs, snRNAs and tRNAs. The miRMaster pipeline denotes any fragment as either tRNA or lncRNA, even though it cannot detect them at full length as we are size selecting for miRNAs as our RNAs of interest[57]. Additionally, we gathered the alignment and mapping information that were produced during the run. For further analysis, we worked with a misclassification rate of 1. We normalized the raw counts with a rpmm-normalization to be able to ensure comparability between the different samples. Next, we filtered the samples and features. We kept only the samples for which more than 2 million reads could be aligned to the mouse genome. Feature filtering was performed by checking if for at least 10% of the samples for at least one group the raw count exceeded or was equal to 5. Subsequently, for the aging cohort, we obtained 844 sequenced samples and 1966 miRNAs (9139 lncRNAs, 30930 piRNAs, 356 rRNAs, 51 scaRNAs, 1542 snoRNAs, 1390 snRNAs and 408 tRNAs) from which we kept 828 samples and 1174 miRNAs (3424 lncRNAs, 605 piRNAs, 194 rRNAs, 27 scaRNAs, 672 snoRNAs, 705 snRNAs and 404 tRNAs). For the acute diet restriction (aDR) and young mouse plasma injection in older mice study (YMP), we added a further quality control step. If we were left with three or fewer samples per brain region after the sample filtering, we additionally discarded all samples of this brain region. This resulted in 118 sequenced and 113 kept samples for the aDR study, and 68 out of 84 samples for the YMP study. For both, we mapped 1966 features and obtained 1345 and 1382 features after filtering, respectively. The microglia dataset consisted of 8 samples, of which all samples and 419 features survived the filtering using an adjusted threshold of 1.8 million reads. All isomiR expressions were also normalized by rpmm-normalization and included the same samples as their corresponding expression tables and the isomiR forms connected to these miRNAs. Therefore, we did not perform a separate filtering for the isomiRs. We called a feature most expressed if its median value over all samples was the highest compared to the other miRNAs. We denoted a feature as expressed in a brain region if at least 10% of the samples of that brain region had a raw count higher than or equal to 5. Human data from ROSMAP[29] was processed with miRMaster 2.0[54] with default settings. We performed the same pre-

processing as explained above up to the feature and samples filtering where we changed the thresholds for the number of aligned reads to 10,000 and the raw count value to 2 what need to be fulfilled for 10% of the samples for either the male or female samples. So, after filtering, we obtained 203 samples and 280 miRNAs.

We use all features for the Uniform Manifold Approximation and Projection (UMAP) calculation, which was done according to McInnes et al.[58] and was colored by information from the metadata. The parameters were selected visually from the results of a pool of linear combinations of default parameters. We prepared the expression matrix for the calculation by standardizing it (z-scoring). As parameters, we chose 0.25 as the minimum distance, as a metric we used the Euclidean, and we started with a random initialization. For the young samples, we used 5 as the size of the local neighborhood, and for the samples from all ages, we used a local neighborhood of 10.

We calculated per brain region, the expressed features (raw count greater than or equal to 5 for at least 10% of the brain region's samples) for each of the seven RNA classes. From those, we determine relative values to the total number of raw counts of all expressed features from all RNA classes of each brain region individually. We display these values in a stacked bar plot per brain region, colored by the RNA classes. Due to the different raw count distributions of the RNA classes, the share of miRNA becomes less prominent compared to the mapping results. By the same approach, we obtain relative values of the RNA classes per brain region on an age-resolved level to gain an overview of the raw count composition over time. Per brain region, we show the relative data points as a scatter plot for each age, colored again by the RNA classes. A third-degree polynomial fitted per RNA class to the respective data points indicates the age-trend of the raw count composition. Using the rpmm-normalized and filtered expression of each RNA class, we calculate the correlations with age of each feature per brain region using Spearman's rank correlation coefficients. Showing the resulting correlation values in a ridgeline plot (like a histogram) joint over the brain regions per RNA class illustrates in which RNA class we obtain the most positive and negative correlations with age.

Box plots for features per brain region resolved by age were calculated. The box borders correspond to the 25th ($Q_1$) and 75th Percentile ($Q_3$), the middle line to the median and whiskers to the minimum (maximum) of the minimum value or $Q_1 - 1.5 \cdot IQR$ (maximum value or the $Q_3 + 1.5 \cdot IQR$), where IQR determines the interquartile range. Solid grey dots in the plot indicate the potential outliers in the data. We fitted a polynomial regression line of degree 2 to the data. For male and female separately, we calculated the Spearman's rank correlation coefficients for each feature with the age per brain region. The corresponding p-values were obtained using a two-sided Spearman's rank correlation test of the null hypothesis that the true correlation is 0. The test statistic is Spearman's $\rho$, and the p-values are computed using an exact permutation test called algorithm AS 89. The Benjamini-Hochberg procedure is used for the adjustment of multiple testing.

To quantify the difference of a brain region to the average brain, we used the coefficient of variation given by the ratio of standard deviation and mean value per feature. We selected the 50 miRNAs yielding the highest value. To differentiate features from the brain average for a particular brain region, we calculated per brain region the median expressions and standardized each feature (z-score). Thus, we called a feature different from the brain average for a specific brain region if the absolute z-score was higher than or equal to 0.5. For the so obtained binarized table, we applied a hierarchical clustering using complete linkage to cluster the binarized expression profiles into four clusters.

Next, for the analysis regarding the sex comparison, Male vs Female, we discarded all samples from the brain region pons due to a lack of samples for older ages in the female case. All Principal Variance Component Analysis[59] (PVCA) in this publication were processed with

properties given by the metadata, all of which could be seen in the PVCA results presented as bar plots, together with two-way interaction terms (containing all combinations of the properties). For clarity, only two-way interaction terms with a non-vanishing observed variance were shown. The residual contains the observed variance that was not covered by the used metadata. While a PVCA is normally used to reveal batch effects within the data, we used it to identify properties suggesting a high impact in the dataset, guiding a further analysis.

In a second investigation, we applied the PVCA to every brain region individually for the properties of age and sex. We applied a DE analysis per brain region for the sex comparison, Male vs Female. We calculated the fold changes for each feature by dividing the geometric median over all male samples by the geometric median over all female samples in every brain region. For any further analysis, we removed the features exhibiting no deregulation (fold change of 1). To obtain the adjusted p-values we performed a two-sided Welch's t-test, which is an adaption of Student's t-test (for simplicity, we always write Student's t-test instead of Welch's t-test), with the corresponding samples and adjusted the obtained p-values using the Benjamini-Hochberg procedure. We called a feature significantly up- or downregulated if the adjusted p-value was smaller than 0.05 and in case of an upregulation (down-) if the fold change was greater than or equal to 1.5 (smaller or equal than 1/1.5). We performed a gene set enrichment analysis for all features. Therefore, we ranked the features with a positive (negative) log2-transformed fold change increasingly (decreasingly) according to their significance (p-value). Subsequently, the enrichment analysis (Gene set enrichment analysis, GSEA) was performed using miEAA 2023[60] with the combined list of ranked features. P-values were calculated using an exact dynamic programming algorithm based on the null distribution of enrichment scores, avoiding the variability of random permutation tests, and were adjusted for multiple testing using the Benjamini–Hochberg procedure, as detailed in Backes et al.[61] and Keller et al.[62].

We called a feature significantly positively or negatively correlated with the age if the adjusted p-value was smaller than 0.05 and positively (negatively) correlated if the correlation value was greater than 0.5 (smaller than −0.5). Additionally, we denoted a feature as unique if it exhibited a significantly positive or significantly negative correlation within only one brain region. If this held for more than one brain region in the same direction, we called it multiple. Subsequently, we determined the deregulated features per brain region by calculating the fold changes analog to above for the comparisons of every older age (12 m, 15 m, 18 m, 21 m and in the case of male add. 26 m, 28 m) versus the control age of 3 m again for male and female separately. Therefore, we used the medians of each group. Again, we determined the p-values with the Student's t-test and adjusted them by using the Benjamini-Hochberg procedure. Analogously to the above for the correlation approach, we introduced the property unique or multiple for features that were significantly up- or downregulated in at least one age comparison within a brain region. If this feature were in only one brain region significantly deregulated, we called it unique, and for more than one, we called it multiple. To close the side-by-side analysis, we combined the male and female data back to one dataset for which we calculated the unique and multiple features for the two approaches as explained above. We call a feature a candidate if it was significantly positively or negatively correlated with age in at least one brain region or significantly up- or downregulated in at least one age comparison for a brain region. If the direction changed between the brain regions, the features were just counted as one candidate.

The human miRNA data from ROSMAP[29] was binned for the age of death in three half-open intervals [71,81), [81, 92), and [92, 103) in years. Afterwards, we perform a DE analysis between the oldest and the youngest group using the Student's t-test and Benjamini-Hochberg to

adjust the p-values for multiple testing. Again, the fold changes were calculated with the geometric means. Additionally, we determined Cohen's d for every feature to obtain a measure for the effect size[63].

Madrer and Soreq[32] introduced a set of miRNAs that are predicted to target cholinergic genes. We intersect this list of miRNAs with the miRNAs of our dataset found to be significantly positively or negatively correlated with age individually for each brain region.

For each significantly age-correlated miRNA, we investigated a 10 kb range on the same and the opposite strand and determined significantly age-correlated miRNAs. Hence, we obtained a tuple of the number of feature candidates on the same and the opposite strand. Summing up the occurrences of these tuples for each significantly age-correlated miRNA within each brain region yields a global overview.

Per brain region, we conduct a gene set enrichment analysis for all features. Therefore, we ranked the features with a positive (negative) Spearman's rank correlation coefficient with age increasingly (decreasingly) according to their significance (p-value of the correlation). We used miEAA 2023[60] with the combined list of ranked features to obtain the GSEA results as explained above.

Using a c-means clustering[64], we clustered the standardized trajectories given by the miRNAs resolved in the brain regions (total: 15 brain regions × 1174 features = 17,610 trajectories). We determined the number of used clusters by visually inspecting the minimum centroid measurements for all cluster numbers from 2 to 200. From the method, we obtained a percentage value for each trajectory and each cluster, containing the probability that the trajectory belongs to this cluster. Note, cluster 52 contained only one trajectory; hence, we excluded it from further evaluations. Using this measure, we assigned each trajectory to the cluster where it exhibits the highest membership and afterwards discarded the ones with a membership lower than 15%. Therefore, each cluster with its specific aging trajectory has a unique composition of tuples of miRNAs and brain regions. Each tuple only occurs once across all clusters. We called a cluster brain region specific if more or equal than 30% of the elements in this cluster belonged to only one brain region, and feature specific if 4 or more occurrences of one feature could be found in the cluster.

miRTargetLink 2.0[43] provided 82 target genes for the feature mmu-miR-155-5p, considering all functional ones. Using the gene data from Hahn et al.[3], we obtained for 66 of the target genes expression values from 809 samples matching to the miRNA samples. We calculated the Spearman's rank correlation coefficients between the target genes and the miRNA 155-5p for every brain region and the corresponding adjusted p-values with the Benjamini-Hochberg procedure. For this part of the analysis, we called a gene significantly negatively correlated if the correlation value was smaller than or equal to 0.3 and the adjusted p-value was smaller than 0.05. Further, we only investigated target genes, and their expression values resolved in the ages for only the brain region for which they were significantly negatively correlated with mmu-miR-155-5p. In the mTOR pathway are 67 genes included[45]. For 63 genes, we calculated the Spearman's rank correlation coefficient using the mRNA data presented by Hahn et al.[3]. As an adjustment method, we used the Benjamini-Hochberg procedure. Analog to above, we calculated the Spearman's rank correlation coefficients for the miRNA mmu-miR-155-5p and 66 from the 80 functionally validated target genes introduced by Hart et al.[44] for each brain region. We consider genes for which at least one brain region exhibits a significant (adjusted p-value < 0.05, Benjamini-Hochberg procedure) correlation value below -0.5.

For data introduced in Keele et al.[46] and published via their web interface Aging B6 Proteomics, we show the intensities for MEF2A in the hippocampus as box plots. The box plots are built like above due to the low number of samples per box, we added black circles visualizing the exact data points within the plots.

The topmost expressed isomiRs were calculated analog to the most expressed miRNAs. The brain region clustering for the standardized isomiR expression data was achieved by hierarchical clustering using complete linkage. The features were ranked based on their expression level.

A visualization of the dataset from Hoye et al.[38] was implemented accordingly to previous publications[39]. The top miRNAs in the microglia dataset were obtained analog to above. The clustering of the features was done by hierarchical clustering using complete linkage. Analogously to the aging cohort, we calculated the top features based on the coefficient of variation and clustered the standardized expression values by samples and features (hierarchical clustering using complete linkage). A DE analysis within the microglia data was done analog as before for the comparison Old versus Young. We selected 5 upregulated (greater than or equal to 1.5) and 5 downregulated features (smaller than or equal to 1/1.5) by their adjusted significance for further analysis. For the aDR and YMP dataset, we obtained a DE analysis analog to the above for the comparison Treatment versus Control and old mice with young plasma versus PBS for all samples and split in the different brain regions. We calculated violin plots for features per brain region resolved by treatment type. The violin displays the density of the data points. Grey dots in the plot represent all data points used for this violin.

The analysis and figures were produced via snakemake pipelines[65] (version 7.18.2) using R (version 4.2.2) and Python (version 3.11.0). Data analysis was done using the package data.table (version 1.14.6), ggrepel (version 0.9.2), reshape2 (version 1.4.4), stringr (version 1.4.1), mfuzz[66] (version 2.58.0) and DESeq2[67] (version 1.38.0). The UMAP calculation in Python was implemented using the package umap-learn[58] (version 0.3.10), numpy (version 1.19), and pandas (version 0.24.2). The heatmaps were created with ComplexHeatmap[68] (version 2.14.0) and circlize (version 0.4.16); all other figures with ggplot2 (version 3.3), gridtext (version 0.1.5), and fontawesome (version 0.4.0).

### Reporting summary

Further information on research design is available in the Nature Portfolio Reporting Summary linked to this article.

## Data availability

The sequencing data generated in this study have been deposited in the NCBI's Gene Expression Omnibus database under accession codes GSE282205 and GSE282207 for the brain aging, invention studies and the microglia data, respectively. We built a web service that offers interactive access to processed bulk-sequencing data from the brain aging cohort, as well as studies on dietary restriction, young plasma injection, and microglia. The web service is accessible via https://ccb-compute2.cs.uni-saarland.de/brainmirmap. Human ROSMAP data is available via Synapse[69]. Source data are provided with this paper.

## Code availability

The code supporting the analysis within this paper is available at GitHub[70].

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

## Acknowledgements
We thank all members of the Wyss-Coray and Meese lab, as well as all members of the Keller lab, for feedback and support. This study is funded by the Saarland University (A.K.), the DAAD (V.W.), NIH Pathway to Independence Award 1K99AG088304-01 (I.H.G), AHA-Allen Brain Health and Cognitive Impairment Cross-Network Collaborative Grants (23BHCICG1188316; N.L.), and the M.J. Fox Foundation (MJFF-021418; A.K. & T.W-C.). Computational resources used within this study were financed through the DFG project 469073465 (A.K.). N.L. is a MAC3 Dementia and Ageing Fellow supported by MAC3 Impact Philanthropies. Special thanks to Phillip Gross (Georgetown University Medical Center) and Ruben Garcia Martin (Centro Nacional de Biotecnología CSIC (CNB), Madrid) for their support.

## Author contributions
Conceptualization: A.K., T.W-C., O.H., V.W.; Methodology: N.L., V.W.; *in-vivo: microglia experiments*: N.L., I.G., A.S., V.W.; Sample collection: O.H., A.F., M.A.; Library Preparation: N.L., A.B., V.W.; Software: A.E.; Resources: O.H., T.W-C.; Data Curation: A.E., V.W.; Writing – Original Draft: A.E., V.W.; Writing – review & Editing: O.H., U.F., A.K., T.W-C.; Visualization: A.E., A.K., V.W.; Supervision: E.M., A.K., T.W-C.; Funding Acquisition: A.K., T.W-C., E.M.

## Funding

## Competing interests
The authors declare no competing interests.
