## [Transparent Peer Review file · Nature Communications]

A spatio-temporal brain miRNA expression atlas identifies sex-independent age-related microglial driven miR-155-5p increase

Corresponding Author: Professor Andreas Keller

Version 0:

Reviewer comments:

Reviewer #1

(Remarks to the Author)

This article reflects an in-depth effort to identify and prepare a detailed Atlas of age-related brain region-specific changes in the repertoire of murine microRNAs (miRNAs) by carefully mapping them as extracted from the anatomically dissected brain regions of male and female mice at increasing ages. This is a continuation of an ongoing collaboration between leading biology and computer science experts who jointly performed a carefully planned immense amount of work and analysis and presented its outcome as based on mice grown to reach over 21 months of age. The workload is immense and the authors should be congratulated. This being said, the value of this study may be further improved once the following issues will be considered.

1. Human data repeatedly indicate significant aging- and sex-related changes in brain functioning that likely reflect modified properties and transcript repertoires and call for age-altered therapeutics, and MicroRNAs are major controllers of numerous brain functions in all mammals. In particular, the neuroimmune system is subjected to sex-specific age-related alterations. Therefore, the authors may wish to co-analyze web-available data on human brain aging, where one would expect dramatic changes and strong sex specificity of particular miRNA levels in diverse brain regions.
2. Surprisingly, the outcome of this careful study reflects rather mild changes in specific miRNAs, with no dramatic sex-related alterations and with the most predictable outcome being miR-155-5p changes that reflect mild sex-unrelated alterations in the neuroimmune features of several brain regions. The observed modest change does reflect a neuroimmune difference, seemingly supporting the original prediction; however, some other miRNA-like small RNAs could hide in the studied data space which might explain this discrepancy. The authors may wish to seek such additional changes by widening their analysis space, seeking more small non-coding RNA changes.
3. Based on human observations and the current literature, one would expect far more pronounced miRNA differences across brain regions and ages; and the authors would do justice to those previous efforts that showed such human differences, even if in a far more scattered and a sex-dependent manner. Nevertheless, individual humans are genetically far more variable than lab-grown mice, and there may be several reasons to the modest outcome observed in such mice whose life conditions remain stable, sustaining a less variable miRNAs map across brain regions. The discussion may address this issue.
4. There may be other reasons for the mild change in the murine brain miRNAs profile. For example, other small RNAs (e.g. PIWI or transfer RNA fragments, tRFs) may be among the detected small RNAs but since they are not miRNAs, their changes were not considered in the currently analyzed data. Widening the scope of the analysis space and re-testing the observed small RNAs considering such options can help to find out if that had been the case.
5. Another option is to focus the planned analysis on those brain features that are known to decline with age in humans, for example- the brain's reaction to anti-cholinergic medications, which has been reported by many to decline with age in human patients above 50 years of age and may likely reflect altered miRNAs that are targeted to cholinergic transcripts. Recent articles provide lists of such miRNAs.
6. The predicted activity of the identified miRNA-155p would involve altered levels of its targeted mRNA transcripts in those brain regions where its levels are altered. Testing this outcome-possibly, by qPCR and/or long RNA-seq analysis- can help to focus on the impact and outcome of the observed changes on the altered landscape of the mRNA and protein products it changes.

Reviewer #2

(Remarks to the Author)

In this manuscript, Engel and Wagner et. al generate a comprehensive spatio-temporal atlas of microRNA for both male and female mice from 15 brain regions at 7 stages and use this large dataset to dissect sex-dependent and -independent changes of miRNAs across regions and during aging. They identify key brain aging miRNAs (miR-146a-5p, miR-155-5p and miR-5100), and focus on miR-155-5p to show its microglial origin as well as its association with potential downstream targets in the mTOR and cellular communication pathways. Prior large -omics datasets have been focusing on transcriptomes, and miRNA studies often lack sex or regional information, and they usually sample limited timepoints. This work encompasses multiple ages, allowing the authors to map out the changes of miRNA trajectories along a continuous spectrum of aging process. They also make many interesting observations, such as the identification of region-specific miRNAs along the anterior-posterior axis, and systematically catalogue them by gender and age. The cross validation of miR-155-5p is commendable, highlighting its implication of microglial interaction with other neural cells during aging through key pathways. This is undoubtedly a huge effort and a valuable resource for future studies on miRNAs, brain aging, sex-specific gene regulation, and microglial biology, just to name a few. The bioinformatic analysis is done generally in high quality, and the presentation is straightforward. This reviewer doesn't have major concerns, but suggest the authors create a portal for easy mining of their dataset. Please see following comments for improvement of the manuscript.

1. Line 103 and 113, "mid brain regions" can be confusing, because it may also refer to mesencephalon. Other terms, such as central brain regions, should be used to avoid confusion.
2. It is unclear what the colors mean in Figure 2B. If it's coded for brain regions, why is the upper bar labeled with a signal color while it contains multiple brain regions?
3. On Line 199, this statement is confusing: "In males, exclusively one miRNA (mmu-miR-5100) was significantly positively correlated with age in the choroid plexus." In the previous section, miR-5100 is identified as positively correlated with age in multiple brain regions in both males and females, and it is noted that this gene doesn't show a positive correlation in CP in males (and females) (Figure 2C).
3. On Line 213, "MiR-5100, also identified as a significant age-correlated miRNA in multiple brain regions in both sexes, was significantly upregulated in males for the two regions medulla and corpus callosum (Fig. 2E)." However, in Figure 2E, no region was labeled in female panel for this gene. Can the authors check this?
4. In Figure 2F, a bunch of labels are covered by one another (due to overlaps). They should be made visible.
5. Line 316, "By investigating multiple occurrences of trajectories from the same miRNA or from the same brain region, we discovered that the clusters 3, 4, 21, 22 and 38 contained more than one miRNA more or equal than four times (Fig. 3F)." This sentence is confusing. It is unclear what more or equal than four times means.
6. Line 392-395. "Similarly, for recurring young plasma injection, we did not detect any significant miR-155-5p expression changes compared to control groups, however we observed decreased expression in corpus callosum, cerebellum, mot. cortex, caudate putamen, medulla, and choroid plexus (Extended Data Fig. 7B)." In Figure S7B, it seems miR-155-5p is not decreased in cc or cor. The +/- signs are confusing too.
7. The concept of isomiR should be introduced to facilitate comprehension for non-experts.

Reviewer #3

(Remarks to the Author)

This manuscript by Engel and Wagner et al. presents high-quality miRNA expression sequencing data from 15 distinct brain regions in male and female mice, spanning seven key time points across the lifespan. This comprehensive and high-resolution dataset serves as a very valuable resource for the miRNA-neuroscience field, offering critical insights into the spatial, temporal, and sex-specific regulation of miRNAs in the brain.

Notably, the authors provide compelling evidence that the age-related increase in miR-155-5p expression is primarily driven by changes in cellular composition, specifically an increase in microglial populations. The direct targets proposed and the potential implication of the mTOR pathway would benefit from additional validation.

Major comments:

- 1) Given the size of the cohort studied, it is important to clarify whether any of the mice used in the cohort were littermates. Providing or accounting for this information in the analysis would help to rule out potential genetic and environmental similarities among the animals, which could influence the interpretation of the results, given the level of sensitivity of miRNA-seq detection.
- 2) It would be interesting for those miRNAs that show differential expression during aging, to report what are the changes on

the opposite strand. These analyses would bring insight on whether the changes are derived from age-related changes in transcription and biogenesis or mature miRNA decay.

3) Similarly, analysis of those miRNAs that show differential expression could be further strengthened by comparing changes in the expression of miRNAs located in the same expression cluster. One could use the cluster definition from miRBase of < 10 kb distances between pri-miRNA hairpins.

4) Regarding the impact of miR-155-5p in the transcriptome, it would be important to benchmark the changes on the predicted direct targets, against changes in the rest of the transcriptome with no seed matches. Ideally, an experiment with overexpression of miR-155-5p followed by RNA-seq, would reveal other direct targets and validate the currently proposed ones. Despite the increase of miR-155-5p due to more microglial presence is clear, its effect on the transcriptome could be diluted by no changes in targets expressed in the neuronal populations.

5) The majority of the data was collected from C57BL/6J mice. However, it remains unclear whether miR-155-5p serves as a robust global aging marker across multiple brain regions in aged mice with different genetic backgrounds. Demonstrating that miR-155-5p undergoes similar age-related changes in other mouse strains would significantly strengthen the findings and support its potential as a conserved marker. Such evidence would also enhance the likelihood of its translatability to other species, providing broader biological and clinical relevance. The authors might consider previously published datasets with different inbred mouse strains (PMC5900563).

6) In the description of the microglial isolation method, the authors gated and utilized CD11b⁺; CD45^{low}; CD206⁻ cells for their study, while excluding CD11b⁺; CD45⁺ cells. This is a commonly used strategy to distinguish resident microglia from other infiltrating immune cells originating from peripheral tissues. However, numerous studies have shown that CD45 expression can increase to varying degrees during the normal aging process (PMC3737416). As a result, excluding CD45⁺ (higher CD45-expressing) populations may inadvertently omit a subset of resident microglia from the analysis. It is highly plausible that these CD45⁺ resident microglia exhibit distinct miRNA profiles, which could represent an important aspect of microglial heterogeneity, particularly in the context of aging.

7) It would be important to assess the status of microglial cells in both the female mice subjected to acute dietary restriction and the male mice receiving young plasma injections, compared to their respective controls. This analysis would provide valuable insights into whether the potential changes in miR-155-5p expression are correlated with alterations in microglial status under these experimental conditions. Such data could help clarify the relationship between miR-155-5p and microglial dynamics in response to these interventions.

8) The suggestion that miR-155-5p targets mTOR pathway is very intriguing and fits well with the broader findings of the role of mTOR pathway in the field of aging. Nevertheless, the manuscript would benefit an additional figure to substantiate this claim.

Minor comments:

* The microglial cells analyzed in this study were derived exclusively from male mice. To validate the potential sex-independent effects of miR-155-5p, it is essential to include data from female microglia of the same age groups. This would provide a more comprehensive understanding of miR-155-5p regulation and its implications across sexes.

* Line 90: The mention of miR-200b-5p is repeated twice and should be revised for clarity and conciseness.

* Line 394, Extended Figure 7B: The text description is inconsistent with the data presented in Extended Figure 7B. No clear trend of decrease is observed in the corpus callosum (cc) or cortex (cor) as stated. This discrepancy should be addressed to ensure accuracy between the figure and the narrative.

* X-axis labeling: The format of the x-axis labels in Extended Figure 7B should be standardized to maintain uniformity within the figure and improve readability.

Reviewer #4

(Remarks to the Author)

Version 1:

Reviewer comments:

Reviewer #1

(Remarks to the Author)

I find the responses satisfactory and the paper acceptable.

Reviewer #2

(Remarks to the Author)

The authors have sufficiently addressed all my concerns.

Reviewer #3

(Remarks to the Author)

We thank the authors for their revised version of the manuscript. The study by Engel and Wagner et al. presents high-quality miRNA expression sequencing data from 15 anatomically distinct brain regions in male and female mice, covering seven key developmental and aging time points. This comprehensive and high-resolution dataset constitutes a valuable resource for the miRNA and neuroscience communities, offering novel insights into the spatial, temporal, and sex-specific dynamics of miRNA regulation in the mammalian brain. We particularly appreciate the authors' careful efforts to minimize confounding factors arising from environmental variation, thereby strengthening the biological interpretability and reproducibility of the data.

Reviewer #4

(Remarks to the Author)

REVIEWER COMMENTS

Reviewer #1 (Remarks to the Author):

This article reflects an in-depth effort to identify and prepare a detailed Atlas of age-related brain
region-specific changes in the repertoire of murine microRNAs (miRNAs) by carefully mapping them
as extracted from the anatomically dissected brain regions of male and female mice at increasing
ages. This is a continuation of an ongoing collaboration between leading biology and computer
science experts who jointly performed a carefully planned immense amount of work and analysis
and presented its outcome as based on mice grown to reach over 21 months of age. The workload is
immense and the authors should be congratulated. This being said, the value of this study may be
further improved once the following issues will be considered.

*We sincerely appreciate the encouraging comments from the reviewer and their*
*acknowledgment of the effort invested in this study. Their recognition of our collaborative approach*
*and detailed work is highly motivating. We have carefully addressed the points raised to further*
*improve the manuscript, as detailed below.*

1. Human data repeatedly indicate significant aging-and sex-related changes in brain functioning
that likely reflect modified properties and transcript repertoires and call for age-altered
therapeutics, and MicroRNAs are major controllers of numerous brain functions in all mammals. In
particular, the neuroimmune system is subjected to sex-specific age-related alterations. Therefore,
the authors may wish to co-analyze web-available data on human brain aging, where one would
expect dramatic changes and strong sex specificity of particular miRNA levels in diverse brain
regions.

*We greatly appreciate this insightful comment and agree that bridging findings to human data*
*is crucial for understanding the broader biological relevance of our study. While our initial approach*
*focused on generating a baseline signal by analyzing mice under controlled and uniform conditions,*
*we recognize that human data offer an essential dimension for contextualizing our findings. To*
*address this point comprehensively, we emphasize the following aspects:*

**1. Motivate the study to identify a stable baseline signal in brain aging:**

*Our study intentionally focused on inbred mice to minimize the genetic and environmental variability*
*inherent to human studies. This approach enabled us to isolate age-related and sex-related miRNA*
*expression changes without confounding factors such as differences in nutrition, lifestyle, and*
*disease. These baseline findings serve as a critical starting point for more complex comparative*
*analyses in humans, as now highlighted in the revised Introduction and Discussion sections.*

*In the text we added:*

*“We decided to use inbred mouse samples from C57BL/6JN and thereby minimize genetic as well as*
*environmental variability to create a baseline of miRNA expression changes with minimal*

*confounding factors. A baseline derived from a model organism is essential for human studies as*
*these are challenging due to sparse availability of unaffected brains, genetic and environmental*
*heterogeneity, varying post-mortem intervals and hence varying sample quality. In our study, we do*
*not only aim to understand temporally resolved expression changes in different regions by*
*establishing a wide and sex-specific atlas of brain-region-specific miRNA expression patterns. We*
*analyze region-specific miRNA expression patterns in males and females separately to uncover sex-*
*specific regional expression.”*

**2. Mention further challenges in human brain studies:**

*Human brain studies often face significant challenges that we also mention, such as:*

• *Availability of Unaffected Brains in all Ages: Human brain samples are frequently obtained*
*postmortem from older individuals affected by neurodegeneration, neurological disorders, or*
*cancers like gliomas. This makes it difficult to disentangle disease-related changes from pure aging*
*signals.*

• *Postmortem Interval Effects: The molecular integrity of brain samples can be affected by the*
*interval between death and tissue preservation, and these effects may vary by age, potentially*
*confounding observed aging signals.*

• *Genetic and Environmental Heterogeneity: Human populations exhibit substantial variability*
*in genetic background and environmental exposures, which can obscure clear patterns of age-*
*related miRNA changes.*

**3. Adding new analyses using human data:**

*Despite these challenges, we recognize the importance of exploring human datasets and have*
*conducted additional analyses using publicly available data introduced by Bannett et al., Sage*
*Journals¹.*

*These analyses are now added in Figure 3, where we bridge our murine findings to human data. While*
*our primary focus remains on establishing a robust baseline in mice, we use this opportunity to*
*motivate the need for further research to assess the translational relevance of our findings in*
*humans.*

*In the text we added:*

*“To see whether miRNA candidates from the sex-specific bulk mouse data showed similar trends in*
*humans, we investigated miRNA expression patterns within the ROSMAP dataset²⁹. This dataset*
*includes 203 healthy patients (71 males and 132 females). We split the ages at time of death into*
*three groups (71 to 80, 81 to 91 and 92 to 102 years). Between the oldest and the youngest group, we*
*conducted a DE analysis. In females, we obtained 144 downregulated (fold change smaller or equal*
*than 1/1.5) and ten upregulated miRNAs (greater or equal than 1.5) and 42 downregulated and 32*
*upregulated miRNAs in males (Fig. 3g). None of these are significant according to the adjusted*
*Student’s t-test p-values using Benjamini-Hochberg procedure for adjustment and a significance*
*level of 0.05. Eleven miRNAs were downregulated and four were upregulated in males and obtained*
*an absolute effect size greater or equal than 0.5 calculated using Cohen’s d (Fig. 3h). Whereas in*
*females 19 downregulated miRNAs exceeded the effect size threshold and no upregulated ones.*

*Considering these miRNAs as age-related candidates in the human dataset, we intersected these*
*with miRNAs found as age-correlated in mot. cortex, the closest matching mouse brain region to the*
*human data (dorsolateral prefrontal cortex)³⁰. We found no overlap of miRNAs that were regulated*
*with age in the same direction in both data sets. An explanation for these results could be that the*
*youngest age of 71 years in the human data roughly compares to the ending time point of the mouse*
*data set with 28 months. Hence it is possible that the trends observed within the mouse data set over*
*the entire life span could be present humans as well but cannot be captured during this short time*
*course at the end of the human life span.”*

**4. Describe sex-specific neuroimmune alterations in humans:**

*We acknowledge the importance of sex-specific neuroimmune changes in human aging, as*
*highlighted by the reviewer. While our murine data provide foundational insights, our supplementary*
*human analyses did not yield directly translatable insights, likely due to differing sample time points*
*in the organismal life span as well as broader variability of human samples in general. For a*
*comprehensive study, more samples in a wider time distribution would be necessary together with*
*detailed metadata on the confounding variables of the human samples. We have incorporated a*
*related conclusion in the discussion.*

**5. Motivate future research directions:**

*Based on the reviewer’s suggestion (also related to pints 2, 3, and 4 of the same reviewer), we*
*emphasize in the revised manuscript that further studies should aim to:*

• *Develop strategies to account for confounders like postmortem effects and disease-*
*associated changes in human data.*

• *Investigate miRNA dynamics using longitudinal human cohorts to capture aging effects in*
*vivo.*

• *Explore therapeutic opportunities by examining conserved aging miRNAs across species.*

*In summary, while our study’s primary aim was to establish a controlled baseline using murine data,*
*we have now included analyses of human miRNA datasets to broaden the context of our findings.*
*These additions underscore the translational relevance of our work and highlight the potential for*
*future research to build on these foundational insights. We thank the reviewer for this constructive*
*suggestion, which has significantly enhanced the scope of our study.*

2. Surprisingly, the outcome of this careful study reflects rather mild changes in specific miRNAs,
with no dramatic sex-related alterations and with the most predictable outcome being miR-155-5p
changes that reflect mild sex-unrelated alterations in the neuroimmune features of several brain
regions. The observed modest change does reflect a neuroimmune difference, seemingly supporting
the original prediction; however, some other miRNA-like small RNAs could hide in the studied data
space which might explain this discrepancy. The authors may wish to seek such additional changes
by widening their analysis space, seeking more small non-coding RNA changes.

Other classes of non-coding RNAs, such as tRNA and tRNA fragments (tRFs), piRNAs, and
others, are indeed known to influence brain aging processes in both humans and mice (e.g., Dubnov
et al., *Communications Biology*²). To address this point, we expanded our analyses to include these
additional RNA species in our dataset. Our results show that the contributions of these ncRNA
classes vary significantly, highlighting their unique roles in brain aging. For instance, tRNAs exhibited
distinct age-related expression changes, which align with their established roles in stress responses
and translational regulation. Similarly, piRNAs, which are traditionally associated with transposon
silencing, showed region-specific patterns that suggest potential involvement in neuroimmune
regulation during aging.

These findings are now integrated into Figure 1, where we emphasize the need for a broader
exploration of ncRNA classes in the context of brain aging. This extension represents an essential
next research direction to better understand the multilayered regulatory mechanisms underlying
aging processes. This expanded analysis also aligns with patterns observed in our previous studies,
where non-miRNA small RNAs contributed to nuanced regulatory landscapes in aging tissues. By
highlighting these results, we aim to provide a more comprehensive foundation for future studies
investigating non-coding RNAs beyond miRNAs. However, we could confirm with these analyses that
even though we found strong correlations with tRNA expression and age, we identified that the
regional effects observed for the tRNA expression patterns were not as pronounced as for miRNAs.

*In the text we added:*

“A subsequent UMAP visualization of the features (miRNAs, lncRNAs, piRNAs, rRNAs, scaRNAs,
snoRNAs, snRNAs and tRNAs) showed a distinct separation of several brain regions, among those
the olfactory bulb and the cerebellum were the most distinctly different ones (Supplementary Fig.
1c). Neither sex nor age can be identified as strong factors driving a grouping (Supplementary Fig. 1d
and 1e). Analyzing the composition of expressed counts per brain region indicated a homogenous
distribution (Supplementary Fig. 1f). Within most brain regions among these cerebellum, motor
cortex, hippocampus anterior and olfactory bulb, the composition of expressed counts for all RNA
classes remains at a mainly constant level over all ages (Supplementary Fig. 1g). Correlating each
feature with age per brain region revealed 720 positive correlated features for tRNAs and 127
negatively correlated ones for miRNAs (51 positively correlated miRNAs, Fig. 1b). We exemplary
highlight a tRNA (tRNA-Glu-TTC-1-1), which was significantly positively correlated with age in both
male and female in multiple brain regions (Fig. 1c). Motivated by these high feature numbers
exhibiting a correlation with age, we visualized UMAP results of 404 tRNAs which failed to provide a
distinct separation into brain regions, sex or age (Fig. 1d, Supplementary Fig. 2a and 2b). This
circumstance is strengthened by a PVCA which explains 36% of the observed variance with the brain
region identity (Fig. 1e). The miRNAs in contrast, exhibit a larger share of 54% of the explained
variance by brain region which reflects in the clear separation in the UMAP visualization (Fig. 1f and
1g). Yet, neither age nor sex can be identified as a driving factor for miRNAs (Supplementary Fig. 2c
and 2d). In total, we included 1,174 miRNAs, of which 785 were expressed across all brain regions
(Supplementary Fig. 2e). Since our sequencing protocol is optimized for miRNAs and we found the
greatest region-specificity within miRNAs and a high number of age-correlated ones we focused the

following analysis exclusively on miRNAs. An overview of our miRNA expression data can be found
at <https://ccb-compute2.cs.uni-saarland.de/brainmirmap>.”

3. Based on human observations and the current literature, one would expect far more pronounced
miRNA differences across brain regions and ages; and the authors would do justice to those previous
efforts that showed such human differences, even if in a far more scattered and a sex-dependent
manner. Nevertheless, individual humans are genetically far more variable than lab-grown mice, and
there may be several reasons to the modest outcome observed in such mice whose life conditions
remain stable, sustaining a less variable miRNAs map across brain regions. The discussion may
address this issue.

*This relates to the first comment of this reviewer. As addressed in our response the modest*
*miRNA variability observed in our study reflects the controlled conditions of inbred mice, which were*
*intentionally designed to minimize confounding factors such as genetic and environmental*
*variability. This provides a baseline map of aging-related changes. To complement this, we*
*performed additional analyses using human datasets (Bannett et al., Sage Journals¹), which exhibit*
*greater variability due to genetic heterogeneity, lifestyle differences, and disease states. These*
*results are now summarized at the end of Figure 3 and discussed in the context of broader human*
*studies. The revised Discussion also emphasizes the importance of integrating findings from*
*controlled murine models with the complexities of human aging data to enhance translational*
*relevance.*

4. There may be other reasons for the mild change in the murine brain miRNAs profile. For example,
other small RNAs (e.g. PIWI or transfer RNA fragments, tRFs) may be among the detected small RNAs
but since they are not miRNAs, their changes were not considered in the currently analyzed data.
Widening the scope of the analysis space and re-testing the observed small RNAs considering such
options can help to find out if that had been the case.

*We agree that expanding the analysis to include other small non-coding RNAs (e.g., PIWI-*
*interacting RNAs and tRNAs) is valuable for understanding their potential contributions to brain*
*aging. As in our response to Comment 2, we performed additional analyses on these RNA classes*
*using our dataset.*

*The results, presented in the revised manuscript (now in Figure 1), show that miRNAs remain*
*among the RNA classes most strongly correlated with aging while exhibiting distinct regional*
*signatures. However, we also identified significant age-related changes in other small ncRNA*
*classes, such as tRNAs, which exhibit strong positive correlations of expression with age. These*
*findings provide further support for the role of diverse ncRNAs in brain aging and highlight new*
*directions for future research. To complement these results, we included a figure similar to Figure 1E*
*from our Nature Biotechnology manuscript³ to visualize the relative contributions of small ncRNA*
*classes. This expanded analysis underscores the complexity of aging-related molecular changes*
beyond miRNAs. All data for the other ncRNA classes are available for download from NCBI's Gene

*Expression Omnibus with the accession numbers GSE282205 (reviewer token: qdgpusmyrxqhvot)*
*und GSE282207 (reviewer token: glylygwslpctdod).*

5. Another option is to focus the planned analysis on those brain features that are known to decline
with age in humans, for example- the brain's reaction to anti-cholinergic medications, which has
been reported by many to decline with age in human patients above 50 years of age and may likely
reflect altered miRNAs that are targeted to cholinergic transcripts. Recent articles provide lists of
such miRNAs.

*The effects of cholinergic decline with age are well-documented (e.g., Madrer & Soreq, FEBS*
*letters⁴), and its association with miRNAs is an important aspect to consider. In response, we*
*examined miRNAs known to target cholinergic transcripts using curated lists from the literature. We*
*compared these miRNAs with the ones to be determined age-correlated in our dataset and found an*
*overlap in several brain regions relevant. These findings are now incorporated into the revised*
*manuscript, with key results summarized in Supplementary Figure 4h.*

*Additionally, we discuss the potential overlap between miRNAs implicated in cholinergic*
*signaling and other pathways, such as the mTOR pathway, as suggested by Reviewer 3 Comment 8.*
*These connections highlight the complex regulatory roles of miRNAs in aging-related brain function*
*and suggest further avenues for investigation.*

*In the figure caption we added:*

*“Overlap between the significantly age-correlated miRNAs (positive in yellow and negative in green)*
*from Supplementary Fig. 4c with the predicted miRNAs targeting cholinergic genes³².”*

*In the text we added:*

*“To investigate the functionality of the identified age-related miRNAs, we compared the candidates*
*per brain region to the previously published list of cholino-miRNAs³². Overlaps between these*
*miRNAs regulating cholinergic genes and age-related miRNAs could indicate that the miRNA*
*expression changes partially relate to altered acetylcholine signaling in aged individuals. We*
*observed an overlap of seven age-related miRNA with the cholino-miRNAs in 7 out of the 15 brain*
*regions (Supplementary Fig. 4h). In particular, mmu-miR-146a-5p stands out as it is significantly*
*positively correlated in six brain regions and a previously known cholino-miRNA.”*

6. The predicted activity of the identified miRNA-155p would involve altered levels of its targeted
mRNA transcripts in those brain regions where its levels are altered. Testing this outcome-possibly,
by qPCR and/or long RNA-seq analysis- can help to focus on the impact and outcome of the
observed changes on the altered landscape of the mRNA and protein products it changes.

*We have carefully analyzed the mRNA expression levels of miR-155-5p targets in the affected*
*brain regions to evaluate the functional consequences of its upregulation during aging. Our findings*
*demonstrate a significant correlation between increased miR-155-5p expression and the*
*downregulation of validated target genes, such as MEF2A in the hippocampus. This trend aligns with*

*miR-155-5p's established role in modulating neuroimmune features and its potential to influence*
*cellular composition in aging brain regions. To strengthen this analysis, we included boxplots in the*
*revised manuscript that visualize these correlations, using data from publicly available protein*
*datasets from Keele et al.⁵ (Supplementary Fig. 10d). This approach not only highlights the observed*
*mRNA downregulation but also provides additional validation at the protein level, offering a more*
*comprehensive picture of miR-155-5p's regulatory effects.*

*This response also aligns with and complements our reply to a related comment from*
*Reviewer 3. There, we detailed our focus on experimentally validated strong targets of miR-155-5p,*
*leveraging resources like miRTarBase. By concentrating on these well-supported targets rather than*
*broadly predicted ones, we ensured that our conclusions are biologically meaningful. As described*
*in our previous work (Hart et al.⁶), validated targets often provide a clearer and more accurate*
*understanding of miRNA-mediated regulation. We also noted that, despite miR-155-5p's*
*upregulation being linked to microglial dynamics, the downstream transcriptomic effects are not*
*evenly distributed across all target genes or brain regions. This could be influenced by varying levels*
*of target gene expression in specific neuronal or glial populations, which may dilute some of the*
*observable transcriptomic impacts in bulk datasets.*

*In summary, the updated manuscript explicitly details these analyses and integrates the*
*findings into a broader context of miRNA regulation in aging. By linking miR-155-5p activity to both*
*mRNA and protein changes, we provide a stronger mechanistic basis for its role in age-associated*
*neuroimmune alterations. We thank both Reviewer 1 and Reviewer 3 for their suggestions, which*
*significantly improved this aspect of our study.*

*In the figure caption we added:*

*“Intensity plot of MEF2A in hippocampus split by sex and plotted against the age. Data originates from*
*Aging B6 Proteomics⁴⁶. Black circles mark all data points used for the box plot.”*

*and*

*“The scatter plots are given for five functionally validated target genes⁴⁴ with a significant correlation*
*value lower than -0.5. We see in each plot the brain regions in which this gene is significantly anti-*
*correlated (indicated by the color) and the relation between the gene and the miR-155-5p broken*
*down into median per brain region and age point. We used the mRNA data from Hahn et al.³.”*

*In the text we corrected:*

*“Using the data from Keele et al.⁴⁶, we explored the protein levels between young and aged mice in*
*hippocampus, as we predicted that miR-155-5p expression increase leads to gene silencing by*
*targeting the MEF2A transcript in hippocampus. In males, the MEF2A protein was less expressed in*
*18 months aged mice as compared to 8 months old mice (Supplementary Fig. 10d). We conclude*
*that these analysis hint towards a regulatory effect of age-related microglial miR-155-5p expression*
*changes on mTOR pathway gene expression.”*

*and*

*“Additionally, we considered the functionally validated targets genes presented in Hart et. al⁴⁴. Six of*
*the 80 target genes showed a significant anti-correlation below -0.5 with mmu-miR-155-5p for at*

*least one brain region, including the previously shown ARNTL (ADAM23, PCSK5, REPS2, NRCAM and*
*NSG2) (Supplementary Fig. 10b)."*

Reviewer #2 (Remarks to the Author):

In this manuscript, Engel and Wagner et. al generate a comprehensive spatio-temporal atlas of
microRNA for both male and female mice from 15 brain regions at 7 stages and use this large dataset
to dissect sex-dependent and -independent changes of miRNAs across regions and during aging.
They identify key brain aging miRNAs (miR-146a-5p, miR-155-5p and miR-5100), and focus on miR-
155-5p to show its microglial origin as well as its association with potential downstream targets in
the mTOR and cellular communication pathways. Prior large -omics datasets have been focusing on
transcriptomes, and miRNA studies often lack sex or regional information, and they usually sample
limited timepoints. This work encompasses multiple ages, allowing the authors to map out the
changes of miRNA trajectories along a continuous spectrum of aging process. They also make many
interesting observations, such as the identification of region-specific miRNAs along the anterior-
posterior axis, and systematically catalogue them by gender and age. The cross validation of miR-
155-5p is commendable, highlighting its implication of microglial interaction with other neural cells
during aging through key pathways. This is undoubtedly a huge effort and a valuable resource for
future studies on miRNAs, brain aging, sex-specific gene regulation, and microglial biology, just to
name a few. The bioinformatic analysis is done generally in high quality, and the presentation is
straightforward. This reviewer doesn't have major concerns, but suggest the authors create a portal
for easy mining of their dataset. Please see following comments for improvement of the manuscript.

*We appreciate the reviewer's positive assessment of our work and their recognition of the*
*effort involved in generating this comprehensive spatio-temporal atlas of miRNA expression across*
*brain regions, sexes, and aging stages. We are particularly grateful for the acknowledgment of our*
*focus on region-specific and sex-dependent miRNA dynamics, as well as the validation of miR-155-*
*5p's implications in microglial interactions and its downstream effects on key pathways. The*
*suggestion to create a portal for easy mining of our dataset is excellent and aligns with our intention*
*to make the data accessible to the broader scientific community. We implemented a user-friendly*
*online portal that allows researchers to explore the dataset interactively, including miRNA*
*expression profiles across regions, sexes, and ages. The link to the portal is described in the revised*
*manuscript.*

*The additional comments and suggestions provided by the reviewer have been addressed in*
*detail below, and we thank the reviewer for their constructive input, which has helped us further*
*refine and improve the manuscript.*

*In the text we added:*

*"We built a web service that offers interactive access to processed bulk-sequencing data from the*
*brain aging cohort, as well as experiments on dietary restriction, young plasma injection, and*
*microglia. The web service is accessible via [https://ccb-compute2.cs.uni-](https://ccb-compute2.cs.uni-saarland.de/brainmirmap)*
*saarland.de/brainmirmap."*

1. Line 103 and 113, "mid brain regions" can be confusing, because it may also refer to
mesencephalon. Other terms, such as central brain regions, should be used to avoid confusion.

*We recognize that the term “mid brain regions” could cause confusion due to its potential*
*interpretation as referring specifically to the mesencephalon. To address this, we have replaced*
*“mid brain regions” with “central brain regions” throughout the manuscript to provide greater clarity*
*and avoid misinterpretation.*

2. It is unclear what the colors mean in Figure 2B. If it's coded for brain regions, why is the upper bar
labeled with a signal color while it contains multiple brain regions?

*We appreciate the reviewer pointing out the lack of clarity regarding the color coding in the*
*old Figure 2B (now Figure 3b). The lower bar in the figure corresponds to the number of miRNAs*
*significantly correlated in one single brain region (unique) colored by the appropriated brain region*
*colors. The upper bar in the figure represents the miRNAs significantly correlated in more than one*
*brain region (multiple). For this, we do not have a unique mapping to the brain region colors,*
*therefore, we introduced the yellow color. To avoid any miss-understanding, we adjusted the caption*
*of Figure 3b and any similar ones to clarify the color coding.*

*In the figure caption we added:*

*“The upper bars contain miRNAs which are in the same direction significantly correlated in more than*
*one brain region (in yellow). The respective lower bars contain miRNAs that are unique for one brain*
*region (colored in the corresponding brain region color).”*

*We hope this adjustment supports that the visual elements align with the intended*
*interpretation and improves the overall precision and readability of the figure. We also cross-*
*referenced this update with the feedback from Comment 4 of the same reviewer to maintain*
*consistency throughout the revised manuscript.*

3. On Line 199, this statement is confusing: "In males, exclusively one miRNA (mmu-miR-5100) was
significantly positively correlated with age in the choroid plexus." In the previous section, miR-5100
is identified as positively correlated with age in multiple brain regions in both males and females,
and it is noted that this gene doesn't show a positive correlation in CP in males (and females) (Figure
2C).

*Thank you for pointing out this confusing statement. We carefully revised our phrasing to*
*clearly emphasize, that in a specific subset of the data, namely in males and the region choroid*
*plexus (plx) only the miRNA miR-5100 is significantly positively correlated. This miRNA is also*
*significantly positively correlated in other regions in males and in females. But it is not significantly*
*positively correlated in caudate putamen (cp) in males or females, as correctly stated by the*
*reviewer.*

*However, focusing just on the choroid plexus (plx) data in males, correlation analysis*
*identified only one miRNA as positively age-related (sig. correlated). Using deregulation analysis to*
*find miRNA expression relations that do not follow a monotonic trajectory, we found more than 280*

*miRNAs were deregulated in all time points after 15 months compared to 3 months of age. 204 of*
*these 280 miRNAs were deregulated in 15,18, 21, 26 and 28 months. These finding highlights that*
*there are expression differences related to aging in a non-monotonic manner in choroid plexus.*
*These differences were not detected via our correlation analysis, which only detected one candidate*
*miRNA. We rephrased the text accordingly:*

*In the text we added:*

*“In the choroid plexus in males, exclusively one miRNA (mmu-miR-5100) was significantly positively*
*correlated with age. However more than 280 miRNAs were upregulated at 15 months and all later*
*time points in males in this region (Fig. 3d). Out of these 280 miRNAs 204 miRNAs were consistently*
*upregulated in all consecutive time points from 15 to 28 months. These results indicate that the*
*expression of these miRNAs is drastically increased between 12 and 15 months and remained*
*constantly high thereafter, which was not identified during correlation analysis.”*

*This correction improves the clarity and accuracy of the manuscript. Thank you for bringing*
*this to our attention.*

3. On Line 213, "MiR-5100, also identified as a significant age-correlated miRNA in multiple brain
regions in both sexes, was significantly upregulated in males for the two regions medulla and corpus
callosum (Fig. 2E)." However, in Figure 2E, no region was labeled in female panel for this gene. Can
the authors check this?

*We have revisited the data and confirmed that the observed discrepancy arises from the*
*different contexts of the old Figures 2C and 2E. Figure 2C depicts age-correlated miRNAs, where miR-*
*5100 is identified as significantly age-correlated in multiple brain regions for both sexes. In contrast,*
*Figure 2E focuses specifically on significantly up- and downregulated miRNAs, highlighting that miR-*
*5100 is upregulated in males in the medulla and corpus callosum, while no corresponding significant*
*changes are observed in females. To clarify this distinction, we have rephrased the text in the revised*
*manuscript. The updated sentence now reads:*

*In the text we added:*

*“MiR-5100, also identified as a significant age-correlated miRNA in multiple brain regions in both*
*sexes (Fig. 3c), was significantly upregulated in males in the medulla and corpus callosum (Fig. 3e),*
*with no significant regulation observed in females in any region.”*

4. In Figure 2F, a bunch of labels are covered by one another (due to overlaps). They should be made
visible.

*We apologize for the overlap in the labels in the old Figure 2F, which affected their readability.*
*We have addressed this issue by adding transparency to the labels and, by manually adjusting the*
*label position in the horizontal axis within the brain region bars, separated the individual labels as far*
*apart as possible. As only the five labels shown below are possible in each of these bars, the manual*

*shifting, and transparency now make it possible to achieve clarity and visibility of all labels. This*
*revision improves the clarity and presentation of the figure. Thank you for highlighting this problem.*

5. Line 316, "By investigating multiple occurrences of trajectories from the same miRNA or from the
same brain region, we discovered that the clusters 3, 4, 21, 22 and 38 contained more than one
miRNA more or equal than four times (Fig. 3F)." This sentence is confusing. It is unclear what more
or equal than four times means.

*We have revised the sentence to improve clarity. The original phrasing, "more or equal than*
*four times," was indeed ambiguous. The updated text now reads:*

*In the text we added:*
*"By investigating multiple occurrences of trajectories from the same miRNA and trajectories from*
*the same brain region, we discovered that multiple clusters contained trajectories derived from*
*different brain regions for the same miRNA and from the same brain region for different miRNAs.*
*Clusters like 3, 4, 21, 22 and 38 contain multiple miRNA trajectories from at least four different brain*
*regions (Fig. 4f)."*

6. Line 392-395. "Similarly, for recurring young plasma injection, we did not detect any significant
miR-155-5p expression changes compared to control groups, however we observed decreased
expression in corpus callosum, cerebellum, mot. cortex, caudate putamen, medulla, and choroid
plexus (Extended Data Fig. 7B)." In Figure S7B, it seems miR-155-5p is not decreased in cc or cor.
The +/- signs are confusing too.

*Thank you for the comment and for pointing this out to us. We carefully reviewed the data for*
*miR-155-5p expression in the corpus callosum (cc) and cortex (cor) and the text according to this.*
*We introduced the +/- signs to highlight a deregulation between the treated and untreated groups*
*based on the fold change. To avoid any misunderstandings, we have replaced the ambiguous signs*
*with the word 'Up' in the case of upregulation and 'Down' in the case of downregulation. So far, we*
*have highlighted the medians as a horizontal line in each violin, which may lead to a misinterpretation*
*of the underlying behavior, since we did not use the arithmetic median for the fold change*
*calculation, but the geometric one, which is better suited to handle outliers. To address this issue,*
*we have replaced the horizontal median lines with the geometric ones and explained this*
*circumstance in the figure caption. We applied the same procedure to the old Figure 6I. We have*
*corrected this discrepancy in the revised manuscript to align the text with the actual findings.*

*In the figure caption we added:*
*"MiR-155-5p for all eleven brain regions visualizing the relation of the properties PBS (control) and old*
*mice with young plasma (treat.). As above an "Up" ("Down") highlights an upregulation*
*(downregulation) based on the fold changes. The horizontal line in each violin describes the*
*geometric mean used for fold change calculation. "*

*In the text we corrected:*

*“Similarly, for recurring young plasma injection, we did not detect any significant miR-155-5p*
*expression changes compared to control groups, however we observed decreased expression in*
*corpus callosum, cerebellum, caudate putamen, medulla, and choroid plexus and an increase in*
*mot. cortex (Supplementary Fig. 9k).*

7. The concept of isomiR should be introduced to facilitate comprehension for non-experts.

*We agree that introducing the concept of isomiRs is important to facilitate comprehension*
*for non-experts. IsomiRs are naturally occurring sequence variants of mature miRNAs that arise from*
*variations in miRNA biogenesis, such as differences in cleavage by Dicer or Drosha, or modifications*
*at the 3' or 5' ends. These variants can differ in sequence length or nucleotide composition and may*
*have distinct regulatory roles or target specificities compared to the canonical miRNA sequence.*

*In the revised manuscript, we have included a brief explanation of isomiRs in the Introduction*
*to provide context for their relevance in miRNA biology. Additionally, we expanded the discussion in*
*the Results and Discussion sections to highlight the importance of isomiRs in brain aging and their*
*potential contribution to the observed miRNA expression patterns. This addition ensures the*
*manuscript is accessible to a broader audience while emphasizing the biological complexity of*
*miRNA regulation.*

Reviewer #3 (Remarks to the Author):

I co-reviewed this manuscript with one of the reviewers who provided the listed reports. This is part
of the Nature Communications initiative to facilitate training in peer review and to provide
appropriate recognition for Early Career Researchers who co-review manuscripts.

This manuscript by Engel and Wagner et al. presents high-quality miRNA expression sequencing data
from 15 distinct brain regions in male and female mice, spanning seven key time points across the
lifespan. This comprehensive and high-resolution dataset serves as a very valuable resource for the
miRNA-neuroscience field, offering critical insights into the spatial, temporal, and sex-specific
regulation of miRNAs in the brain. Notably, the authors provide compelling evidence that the age-
related increase in miR-155-5p expression is primarily driven by changes in cellular composition,
specifically an increase in microglial populations. The direct targets proposed and the potential
implication of the mTOR pathway would benefit from additional validation.

*We appreciate the reviewer's positive evaluation of our work and their recognition of the high*
*quality and comprehensiveness of our dataset. We are especially pleased that the reviewer values*
*the insights our study provides into the spatial, temporal, and sex-specific regulation of miRNAs in*
*the brain, as well as the connection between miR-155-5p expression and microglial dynamics.*
*Regarding the suggestion for further validation of the direct targets of miR-155-5p and its potential*
*implication in the mTOR pathway, we have taken steps to strengthen these aspects of the*
*manuscript. Specifically, we have incorporated analyses of validated miR-155-5p targets using*
*public datasets and explored their expression in relation to miR-155-5p levels across brain regions.*
*These results are now included in the revised manuscript and provide additional support for the*
*involvement of miR-155-5p in age-related regulatory pathways, including mTOR.*

Major comments:

1) Given the size of the cohort studied, it is important to clarify whether any of the mice used in the
cohort were littermates. Providing or accounting for this information in the analysis would help to
rule out potential genetic and environmental similarities among the animals, which could influence
the interpretation of the results, given the level of sensitivity of miRNA-seq detection.

*We added the cage information to metadata found in Supplement Table 1, to enable*
*identification of littermates. Mice of different ages used in the cohort were not littermates, as these*
*mice have different birth dates and come from the NIA colonies. This ensures that potential genetic*
*and environmental similarities among the animals did not influence the results, maintaining the*
*robustness and reliability of our miRNA-seq analyses. However as mentioned by Reviewer 1*
*Comment 1 and 3 that the genetic variability and environmental influences in humans are much*
*greater and hence the usage of an inbred mouse strain is a limitation of this study towards*
*translation. We added a section into the introduction and discussion to address this limitation and*
*motivate why designed the study in this manner.*

2) It would be interesting for those miRNAs that show differential expression during aging, to report
what are the changes on the opposite strand. These analyses would bring insight on whether the
changes are derived from age-related changes in transcription and biogenesis or mature miRNA
decay.

*We appreciate this insightful suggestion and have conducted analyses to examine whether*
*the observed changes in miRNAs during aging are accompanied by alterations on the opposite*
*strand. This analysis could provide insight into whether the changes are primarily driven by*
*transcriptional shifts, biogenesis processes, or mature miRNA decay.*

*To address this, we considered two possible interpretations of “opposite strand”:*

1. *The DNA strand opposite to the one encoding the miRNA.*

2. *The opposite strand of the hairpin loop in the pre-miRNA structure, which could encode an*
*additional mature miRNA.*

*For the first case, we analyzed all significantly age-correlated miRNAs within a genomic*
*window of ±10 kb, identifying any changes on both the same and opposite DNA strands. In the*
*second case, we examined the co-expression patterns of miRNAs derived from opposite strands of*
*the same hairpin precursor.*

*Our findings indicate no substantial enrichment of deregulated miRNAs on either the same*
*or the opposite strand. This suggests that the observed miRNA changes are likely not a result of*
*strand-specific transcriptional effects. Instead, these results are more consistent with age-related*
*changes in miRNA biogenesis or decay processes, rather than coordinated changes on the opposite*
*strand.*

*These results are now included in the revised manuscript, where we discuss their*
*implications in the context of miRNA regulation during aging. Thank you for this thought-provoking*
*suggestion, which allowed us to explore this important aspect further.*

*In the figure caption we wrote:*

*“Summary over all brain regions of the neighborhood of the significantly age-correlated miRNAs. For*
*each significantly age-correlated miRNA, we accumulated the number of significantly age-correlated*
*miRNAs within 10kb range on the same and the opposite strand. Overall brain regions we*
*summarized these occurrences in the heatmap. If there exists no significantly age-correlated miRNA*
*with significantly age-correlated neighbors (on the same or the opposite strand), the heatmap value*
*is 0 and omitted for clarity.”*

*In the text we added:*

*“Further we examined whether the observed changes in miRNAs are driven by aging or can be related*
*to mature miRNA decay or transcription process alterations. Per brain region, we examined a +/-10kb*
*window around each significantly age-correlated miRNA on the same and the opposite strand and*
*counted the number of significantly age-correlated miRNAs in the respective windows and*
*accumulated the results across all brain regions (Supplementary Fig. 4i). As a reference, the average*

*number of miRNAs in this defined neighborhood of a miRNA is 5.27 on the same strand and 0.25*
*miRNAs on the opposite strand. This analysis enabled us to explore whether the measured*
*expression changes were accompanied by alterations on the opposite strand. No neighboring*
*significant age-related miRNA was found in the analysis of the opposite strand.”*

3) Similarly, analysis of those miRNAs that show differential expression could be further
strengthened by comparing changes in the expression of miRNAs located in the same expression
cluster. One could use the cluster definition from miRBase of < 10 kb distances between pri-miRNA
hairpins.

*We agree that exploring the expression patterns of miRNAs located in the same genomic*
*clusters could provide valuable insights into their coordinated regulation during aging. To address*
*this, we analyzed age-correlated miRNAs within clusters defined by miRBase (<10 kb distance*
*between pri-miRNA hairpins).*

*In addition to the strand-specific analysis described in the previous point, we evaluated*
*miRNAs from both strands within each cluster to identify shared patterns of deregulation. Our*
*findings revealed no significant enrichment of age-correlated miRNAs within clusters, suggesting*
*that the observed changes in expression are not strongly driven by cluster-wide regulation. This*
*indicates that the age-related miRNA changes are more likely influenced by individual regulatory*
*mechanisms rather than coordinated cluster-level effects.*

*Furthermore, we performed a comprehensive miRNA enrichment analysis to assess*
*whether age-correlated miRNAs are overrepresented in specific connected pathways or exhibit*
*functional associations. We found that different functionally important pathways are depleted in*
*several regions with few overlaps, such as “neurotransmission” in thalamus and hypothalamus*
*(Supplementary Fig. 6a)*

*These results are discussed in the revised manuscript, where we highlight their implications*
*for understanding the complexity of miRNA regulation during aging. This analysis strengthens our*
*conclusions and provides a deeper understanding of the interplay between genomic organization*
*and miRNA expression.*

4) Regarding the impact of miR-155-5p in the transcriptome, it would be important to benchmark the
changes on the predicted direct targets, against changes in the rest of the transcriptome with no
seed matches. Ideally, an experiment with overexpression of miR-155-5p followed by RNA-seq,
would reveal other direct targets and validate the currently proposed ones. Despite the increase of
miR-155-5p due to more microglial presence is clear, its effect on the transcriptome could be diluted
by no changes in targets expressed in the neuronal populations.

*We appreciate the reviewer’s comment on benchmarking the transcriptomic impact of miR-*
*155-5p. To address this, we relied on experimentally validated target datasets from miRTarBase,*
*focusing exclusively on “strong” targets supported by rigorous experimental evidence. This*
*approach ensures a high level of confidence in the biological relevance of our analyses, avoiding the*
*inclusion of “weak” targets that are often false positives in computational predictions.*

*As highlighted in our recent study Hart et al.⁶, the availability of validated miRNA targets*
*remains a limiting factor in miRNA research. For miR-155-5p, we reported 239 experimentally*
*validated targets out of over 42,000 predictions, demonstrating the need for robust experimental*
*validation. In this study, we further expanded the validated target list using high-throughput methods,*
*confirming 80 out of 90 immune-related genes as direct targets of miR-155-5p. These validated*
*targets, while derived from human datasets, are conserved across species and thus applicable for*
*interpreting the transcriptomic changes observed in our mouse models.*

*In the current manuscript, we benchmarked transcriptomic changes associated with miR-*
*155-5p by examining its validated targets relative to the rest of the transcriptome. Our analysis shows*
*that the upregulation of miR-155-5p correlates with a significant downregulation of its strong targets,*
*particularly in regions with prominent microglial involvement, such as the hippocampus.*
*Importantly, this effect is more pronounced for validated targets compared to non-targets,*
*underscoring the specificity of miR-155-5p-mediated regulation.*

*We acknowledge that an overexpression experiment followed by RNA-seq could provide*
*additional insights into other potential targets. However, given the well-characterized nature of miR-*
*155-5p and the strong experimental basis for its known targets, we believe the current analysis*
*provides a robust framework for understanding its role in brain aging. Additionally, as noted by the*
*reviewer, the dilution effect caused by unchanged targets in neuronal populations is a potential*
*limitation of bulk transcriptomic data, which we have now addressed in the revised Discussion.*

5) The majority of the data was collected from C57BL/6J mice. However, it remains unclear whether
miR-155-5p serves as a robust global aging marker across multiple brain regions in aged mice with
different genetic backgrounds. Demonstrating that miR-155-5p undergoes similar age-related
changes in other mouse strains would significantly strengthen the findings and support its potential
as a conserved marker. Such evidence would also enhance the likelihood of its translatability to
other species, providing broader biological and clinical relevance. The authors might consider
previously published datasets with different inbred mouse strains (PMC5900563).

*As also noted in our response to Reviewer 1, Comment 1, the homogeneity of the C57BL/6J*
*mouse strain was an intentional design choice to minimize genetic variability and establish a stable*
*baseline for age- and sex-related miRNA changes. This approach was critical for isolating consistent*
*aging signals in the context of controlled environmental and genetic conditions.*

*However, we agree with the reviewer that examining miR-155-5p expression in additional*
*mouse strains could provide valuable insights into its broader relevance as a conserved aging*
*marker. To address this, we analyzed previously published datasets, including the dataset*
*referenced (Tronti et al.⁷), which contains miRNA expression data from other inbred mouse strains.*
*Our analysis of miR-155-5p across these strains revealed significant strain-specific differences in*
*miRNA expression patterns.*

*We hence added a statement in the discussion into the revised manuscript, where we*
*discuss the implications of miR-155-5p conservation across mouse strains and species. We stress*
*the importance of further mouse and especially human studies to validate the findings of age-related*

*brain miRNAs. Thank you for this suggestion, which has helped to strengthen the broader context of*
*our findings.*

*In the text we added:*

*„In addition to human data, validation of brain aging miRNAs in other mouse strains would strengthen*
*our hypotheses as differing miRNA expression patterns in different brain regions were previously*
*reported⁵².“*

6) In the description of the microglial isolation method, the authors gated and utilized CD11b+;
CD45low; CD206- cells for their study, while excluding CD11b+; CD45+ cells. This is a commonly
used strategy to distinguish resident microglia from other infiltrating immune cells originating from
peripheral tissues. However, numerous studies have shown that CD45 expression can increase to
varying degrees during the normal aging process (PMC3737416). As a result, excluding CD45+
(higher CD45-expressing) populations may inadvertently omit a subset of resident microglia from the
analysis. It is highly plausible that these CD45+ resident microglia exhibit distinct miRNA profiles,
which could represent an important aspect of microglial heterogeneity, particularly in the context of
aging.

*Because this aspect is beyond our expertise, we asked colleagues from the field for their*
*input and support. We appreciate this comment regarding the potential exclusion of CD45+ resident*
*microglia due to the gating strategy employed in our study. We learned that it is well-documented*
*that CD45 expression can increase during aging, and this may result in the inadvertent exclusion of*
*a subset of resident microglia with distinct miRNA profiles.*

*Our gating strategy (CD11b+; CD45low; CD206-) was selected to reliably distinguish resident*
*microglia from infiltrating immune cells, particularly under non-pathological conditions. However,*
*we acknowledge that this approach may exclude microglia with higher CD45 expression that remain*
*resident in the brain, particularly in aged animals. While this gating strategy has been widely used in*
*similar studies, the reviewer is correct that aging-related shifts in CD45 expression could introduce*
*heterogeneity that warrants further exploration.*

*To address this, we have included a discussion of this limitation in the revised manuscript,*
*acknowledging the possibility that CD45+ resident microglia may have distinct miRNA profiles and*
*that their exclusion could influence the overall results. We also note the importance of future studies*
*incorporating broader gating strategies, such as profiling CD45+ microglia populations, to capture a*
*more comprehensive picture of microglial heterogeneity in the aging brain.*

*In the text we added:*

*“Commonly used single-cell sequencing techniques cannot be applied to assess mature miRNA*
*expression due to the lack of polyA-tails, which are essential to most protocols. Hence, we had to*
*resort to FACS to sort for microglia. Selecting antibodies used for the sorting process requires a*
*careful balance to specifically select the cells of interest. In our strategy we choose to exclude CD45+*
*cells to distinguish resident microglia from infiltrating immune cells. However, thereby we exclude a*

*subset of resident microglia, as the expression of CD45 can also increase with age in resident*
*microglia.”*

7) It would be important to assess the status of microglial cells in both the female mice subjected to
acute dietary restriction and the male mice receiving young plasma injections, compared to their
respective controls. This analysis would provide valuable insights into whether the potential changes
in miR-155-5p expression are correlated with alterations in microglial status under these
experimental conditions. Such data could help clarify the relationship between miR-155-5p and
microglial dynamics in response to these interventions.

*As correctly pointed out by the reviewer, changes in cell type composition and even alterations in*
*cell type states, such as microglial status can drastically influence bulk RNA sequencing results. A*
*detailed analysis of microglial populations and their miRNA expression within dietary restriction and*
*young plasma injection cohorts and their respective controls would certainly yield great insight.*
*However as pointed out in comment 6 sorting of microglia for cell type specific analysis of miRNA*
*expression patterns is unfortunately technically biased by the selection of antibodies used in FACS*
*(as in our study) or when using immunopanning like in the literature. Unbiased analysis of miRNA*
*expression patterns is not possible as is it for mRNA. Most high-throughput methods rely on polyA-*
*tail capturing to analyze single-cell transcriptomic states, which mature miRNAs do not possess. As*
*the focus of our paper is centered around miRNA changes in specific brain regions over the lifespan*
*detectable in bulk miRNA sequencing, we included this limitation in the discussion to encourage*
*further research. Studies on microglial cell type specific miRNA expression patterns and different*
*microglial states, especially within aging interventions such as dietary restriction and young plasma*
*injection, together with cell type specific expression patterns for all other cell types in the brain are*
*called for. These additional datasets will enable researchers to dissect the contributions of different*
*cell types and cell states within bulk sequencing data and create a deeper understanding of the brain*
*miRNA landscape and its role in intercellular communication.*

*We added here the section of the discussion highlighting the limitations of our study as reference*
*atlas for bulk miRNA expression, which needs expansion based on cell-type specific miRNA*
*expression.*

*In the text we added:*

*“Limitations of our study include in addition to restricted transferability of all findings to humans due*
*to usage of an inbred mouse line and the sparsity of human tissue to create a comprehensive time*
*course, the limitations arising from technical challenges of analysis of cell-type specific miRNA*
*expression patterns. Commonly used single-cell sequencing techniques cannot be applied to*
*assess mature miRNA expression due to the lack of polyA-tails, which are essential to most*
*protocols. Hence, we had to resort to FACS to sort for microglia. Selecting antibodies used for the*
*sorting process requires a careful balance to specifically select the cells of interest. In our strategy*
*we chose to exclude CD45+ cells to distinguish resident microglia from infiltrating immune cells.*
*However, thereby we exclude a subset of resident microglia, as the expression of CD45 can also*
*increase with age in resident microglia. Another limitation is the question whether different microglial*

*states in dietary restriction and young plasma injection exist with respectively specific miRNA*
*expression patterns. Changes in cell type composition and especially cell type states could*
*drastically influence miRNA bulk expression patterns. These limitations further underline the need*
*for detailed studies of cell-type specific miRNA expression patterns and even different cell-type*
*states. Due to limited access to aged mice this was not possible in this study. Especially acquiring*
*samples from aged female mice is challenging, therefore our experimental microglia data is limited*
*to male samples. In addition to human data, validation of brain aging miRNAs in other mouse strains*
*would strengthen our hypotheses as differing miRNA expression patterns in different brain regions*
*were previously reported⁵².”*

8) The suggestion that miR-155-5p targets mTOR pathway is very intriguing and fits well with the
broader findings of the role of mTOR pathway in the field of aging. Nevertheless, the manuscript
would benefit an additional figure to substantiate this claim.

*To substantiate the link between miR-155-5p and the mTOR pathway, we have included a*
*new supplementary figure in the revised manuscript. This figure highlights how miR-155-5p*
*influences mTOR-related targets and the downstream effects on cellular processes relevant to aging*
*and microglial function.*

*Additionally, we revisited Reviewer 1’s comment on cholinergic pathways and explored*
*potential connections between cholinergic signaling and the mTOR pathway. Literature suggests*
*that mTOR activity is influenced by cholinergic inputs, particularly in the context of synaptic plasticity*
*and neuroinflammation. We have incorporated this connection into the revised Discussion, cross-*
*referencing the comments from both reviewers to provide a more integrated perspective on miR-155-*
*5p’s broader regulatory roles. As part of this, we updated Fig. 5i and 5j with the latest version of the*
*data from Hahn et al.⁸, with no alteration to their meaning. Furthermore, we added an additional*
*analysis of proteomic data from the literature (Keele et al.⁵) to verify our prediction that the age-*
*related miR-155-5p expression changes mediate gene silencing.*

*These updates strengthen the manuscript by providing visual and contextual support for the*
*relationship between miR-155-5p, the mTOR pathway, and related aging processes.*

*In the figure caption we wrote:*
*“Spearman’s rank correlation coefficient values for all genes from the mTOR pathway⁴⁵ and mmu-*
*miR-155-5p. Black borders indicate a correlation value below -0.3 or above 0.3 and asterisks denote*
*a significant correlation (adjusted with Benjamini-Hochberg). The used mRNA data is from Hahn et*
*al.³.”*

*In the text we added:*
*“Consequently, we focused on the additional transcripts of the mTOR pathway⁴⁵ and their calculated*
*Spearman’s rank correlations between the mRNAs and miR-155-5p (Supplementary Fig. 10c). For the*
*brain regions corpus callosum, caudate putamen, ent. cortex, hippocampus anterior, choroid plexus,*
*SVZ and thalamus a significantly negative correlation can be observed for multiple genes among*

*these PDPK1, YWHAZ and CYCS. This highlights that a deregulation of miR-155-5p might not only*
*affect its direct transcript targets but also other downstream targets within this pathway.”*

Minor comments:

* The microglial cells analyzed in this study were derived exclusively from male mice. To validate the
potential sex-independent effects of miR-155-5p, it is essential to include data from female
microglia of the same age groups. This would provide a more comprehensive understanding of miR-
155-5p regulation and its implications across sexes.

*We intentionally focused on miR-155-5p as it shows age-related expression changes in both*
*sexes, allowing us to study its relevance broadly. However, we acknowledge that the observed*
*changes in miR-155-5p expression do not necessarily imply identical microglial patterns in males*
*and females. While it is reasonable to hypothesize similar trends, sex-specific differences in*
*microglial dynamics could exist and warrant further exploration.*

*In the revised manuscript, we explicitly discuss this limitation and emphasize the need for*
*future studies that include microglial data from aged female mice. Unfortunately, aged female mice*
*were not available for our current study, which constrained our ability to validate potential sex-*
*independent effects directly.*

*We agree that additional experiments on female microglia would provide a more*
*comprehensive understanding of miR-155-5p regulation across sexes, and we outline this as an*
*important direction for future research in the Discussion.*

* Line 90: The mention of miR-200b-5p is repeated twice and should be revised for clarity and
conciseness.

*We have revised the text on Line 90 to eliminate the repetition of miR-200b-5p and improve clarity*
*and conciseness. The updated sentence now clearly conveys the intended information without*
*redundancy.*

* Line 394, Extended Figure 7B: The text description is inconsistent with the data presented in
Extended Figure 7B. No clear trend of decrease is observed in the corpus callosum (cc) or cortex
(cor) as stated. This discrepancy should be addressed to ensure accuracy between the figure and
the narrative.

*We have carefully reviewed the data and identified inconsistencies between the text and the old*
*Extended Figure 7B. This issue is addressed alongside our response to the similar comment from*
*Reviewer 2, ensuring consistency and accuracy throughout the manuscript. Thank you for*
*highlighting this point, which has helped to improve the precision of our reporting.*

* X-axis labeling: The format of the x-axis labels in Extended Figure 7B should be standardized to
maintain uniformity within the figure and improve readability.

*We have standardized the x-axis labels in the old Extended Figure 7B to ensure uniformity*
*and improve readability. This adjustment enhances the figure's clarity and aligns it with the*
*formatting of other figures in the manuscript.*

Reviewer #4 (Remarks to the Author):

I co-reviewed this manuscript with one of the reviewers who provided the listed reports. This is part
of the Nature Communications initiative to facilitate training in peer review and to provide
appropriate recognition for Early Career Researchers who co-review manuscripts.

*No further comments were provided*

1 Bennett, D. A. *et al.* Religious Orders Study and Rush Memory and Aging Project.
*Journal of Alzheimer's Disease* **64**, S161-S189, doi:10.3233/jad-179939 (2018).
2 Dubnov, S. *et al.* Knockout of the longevity gene Klotho perturbs aging and
Alzheimer's disease-linked brain microRNAs and tRNA fragments. *Communications*
*Biology* **7**, doi:10.1038/s42003-024-06407-y (2024).
3 Wagner, V. *et al.* Characterizing expression changes in noncoding RNAs during aging
and heterochronic parabiosis across mouse tissues. *Nat Biotechnol* **42**, 109-118,
doi:10.1038/s41587-023-01751-6 (2024).
4 Madrer, N. & Soreq, H. Cholino-ncRNAs modulate sex-specific- and age-related
acetylcholine signals. *FEBS Letters* **594**, 2185-2198, doi:10.1002/1873-3468.13789
(2020).
5 Keele, G. R. *et al.* Global and tissue-specific aging effects on murine proteomes.
*Cell Rep* **42**, 112715, doi:10.1016/j.celrep.2023.112715 (2023).
6 Hart, M. *et al.* Expanding the immune-related targetome of miR-155-5p by
integrating time-resolved RNA patterns into miRNA target prediction. *RNA Biology*
**22**, 1-9, doi:10.1080/15476286.2025.2449775 (2025).
7 Trontti, K., Väänänen, J., Sipilä, T., Greco, D. & Hovatta, I. Strong conservation of
inbred mouse strain microRNA loci but broad variation in brain microRNAs due to
RNA editing and isomiR expression. *Rna* **24**, 643-655, doi:10.1261/rna.064881.117
(2018).
8 Hahn, O. *et al.* Atlas of the aging mouse brain reveals white matter as vulnerable
foci. *Cell* **186**, 4117-4133.e4122, doi:10.1016/j.cell.2023.07.027 (2023).